# Minimal Invasive Piezoelectric Osteotomy in Neurosurgery: Technic, Applications, and Clinical Outcomes of a Retrospective Case Series

**DOI:** 10.3390/vetsci7020068

**Published:** 2020-05-22

**Authors:** Alberto Maria Crovace, Sabino Luzzi, Luca Lacitignola, Gerardo Fatone, Alice Giotta Lucifero, Tomaso Vercellotti, Antonio Crovace

**Affiliations:** 1Department of Emergency and Organ Transplantation, University of Bari “Aldo Moro”, 70121 Bari, Italy; luca.lacitignola@uniba.it (L.L.); antonio.crovace@uniba.it (A.C.); 2Neurosurgery Unit, Department of Clinical-Surgical, Diagnostic and Pediatric Sciences, University of Pavia, 27100 Pavia, Italy; alicelucifero@gmail.com; 3Neurosurgery Unit, Department of Surgical Sciences, Fondazione IRCCS Policlinico San Matteo, 27100 Pavia, Italy; 4Department of Veterinary Medicine and Animal Productions, University of Naples Federico II, 80138 Napoli, Italy; gerardo.fatone@unina.it; 5Eastman Dental Institute, University College of London, WC1E 6BT London, UK; tomaso@vercellotti.com

**Keywords:** craniotomy, laminectomy, neurosurgery, osteotomy, piezosurgery

## Abstract

Objective: To report the physical and technical principles, clinical applications, and outcomes of the minimal invasive piezoelectric osteotomy in a consecutive veterinary neurosurgical series. Methods: A series of 292 dogs and 32 cats underwent an osteotomy because a neurosurgical pathology performed with a Mectron Piezosurgery^®^ bone scalpel (Mectron Medical Technology, Genoa, Italy) was retrospectively reviewed. Efficacy, precision, safety, and blood loss were evaluated intraoperatively by two different surgeons, on a case-by-case basis. Postoperative Rx and CT scans were used to assess the selectivity and precision of the osteotomy. A histological study on bony specimens at the osteotomized surface was carried out to evaluate the effects of piezoelectric cutting on the osteocytes and osteoblasts. All the patients underwent a six-months follow-up. A series of illustrative cases was reported. Results: All the osteotomies were clear-cut and precise. A complete sparing of soft and nervous tissues and vasculature was observed. The operative field was blood- and heat-free in all cases. A range of inserts, largely different in shape and length, were allowed to treat deep and difficult-to-reach sites. Two mechanical complications occurred. Average blood loss in dogs’ group was 52, 47, and 56 mL for traumatic, degenerative, and neoplastic lesions, respectively, whereas it was 25 mL for traumatized cats. A fast recovery of functions was observed in most of the treated cases, early on, at the first sixth-month evaluation. Histology on bone flaps showed the presence of live osteocytes and osteoblasts at the osteotomized surface in 92% of cases. Conclusions: Piezosurgery is based on the physical principle of the indirect piezo effect. Piezoelectric osteotomy is selective, effective, and safe in bone cutting during neurosurgical veterinary procedures. It can be considered a minimal invasive technique, as it is able to spare the neighboring soft tissues and neurovascular structures.

## 1. Introduction

Piezoelectric osteotomy is based upon the mechanical effect of ultrasound, which is increased by physical phenomenon of cavitation.

Microvibrations of the ultrasonic frequency are linear in shape and micrometric (60–210 μm). They allow us to selectively cut the bone without significant damage to the underlying soft and neurovascular tissues [1,2,3,4,5,6,7,8,9,10,11]. Oscillation frequency ranges between 25 and 30 kHz, the same frequencies at which soft tissues, dura mater, nerves, and blood vessels oscillate at the same time being spared [8,12,13]. On the other hand, the cavitation phenomenon involves the formation of vapor-filled cavities secondary to the abrupt and rapid changes of pressure in a liquid. The input coming from a high-pressure source causes collapse of these cavities and generates a shock wave, which propagates in the tissue. In turn, the shock wave causes cell detachment from the substrate and significant changes in focal adhesion and biomechanics, ultimately inducing a mechanical cut on the mineralized tissue [14,15,16,17].

In piezoelectric surgery, the changes of pressure are generated by a spatial deformation of the piezoelectric crystals when they are subject to an electric field having a given ultrasonic frequency. This mechanism is also called indirect piezo effect [4,15,18,19]. The resulting oscillations are then amplified and transferred onto an instrument tip. Piezoelectric osteotomy is also considered as a “heat-free” technique.

In fact, although the cavitation effects can increase the temperature of the neighboring tissues, the dissipation of the heat coming from the concomitant irrigation practically leads to the avoidance of any thermal injury. It also promotes the absence of blood and debris in the surgical field [4,5,6,7,8,9,10,11,20].

All of these aspects made piezoelectric surgery an attractive option, in comparison with common rotating instruments, for human and veterinary procedures, everything in the context of the ongoing biotechnological revolution which has involved especially neurosurgery [21,22,23,24,25,26,27,28,29,30,31,32,33,34,35,36]. 

However, the literature counts still few studies about its potential implication and safety in routinely clinical practice.

The purpose of this study is to report the overall results of a consecutive retrospective veterinary series where piezoelectric surgery was employed for neurosurgical procedures. Physical and technical principles, as well as outcomes, are reported. 

## 2. Materials and Methods

The present study was approved by the internal Institutional Review Board. 

Overall data of a consecutive surgical series of patients who underwent a surgical procedure executed with the piezoelectric bone scalpel (Mectron Piezosurgery, Mectron Medical Technology, Genoa, Italy) were retrospectively reviewed.

Inclusion criteria involved dogs and cats which required a neurosurgical procedure comprehending a spinal or skull osteotomy because of a traumatic, degenerative, or neoplastic lesion. Animals older than 10 year of age or harboring more than a single neurosurgical pathology were excluded.

In all spinal cases, the patients underwent to a ventro-dorsal and latero-lateral projection X-ray. A spinal computed tomography (CT) scan and magnetic resonance imaging (MRI) were proposed in all cases but performed only in those where the owner was available. Conversely, a CT scan was instead mandatory in all cranial cases. Images were reviewed and edited on a DICOM imaging workstation (Osirix DICOM Viewer@, Pixmeo, Bernex, Switzerland), which also allowed some measurements.

In spinal cases, X-ray and CT scan were both aimed to study the bony component of the pathology. A 3D rendering of the spinal CT scan was also performed, in all cases, to study in more detail the site and cause of compression. In cranial cases, the 3D CT scan allowed us to better localize the lesion. MRI was paramount to study the neurovascular anatomical structures and soft tissues.

All surgeries were performed by the senior surgeon (A.C.). The anesthesia protocol involved the administration of acepromazine 0.005 mg/kg and morphine (0.3 mg/kg) IM for premedication, propofol 4–5 mg/kg IV for induction, and isoflurane-O2 and fentanyl infusion (0.002–0.005 mg/kg/h) for maintenance. 

The piezoelectric device consisted of a power unit connected to an ergonomic piezoelectric hand-piece (Figure 1a), an integrated irrigation peristaltic module, and different specifically designed titanium nitride coated tips of various shapes (Figure 1b–d). Frequency modulation ranged between 25 and 30 KHz with linear microvibrations between 60 and 120 μm. The instrument’s power was based on specific presets varying on the basis of the tissue density. The presets ranged between 2.8 and 16 W. The irrigation pump was set to continuously deliver 30 mL/min of saline solution in all cases, regardless of the power setting. The irrigation rate of 30 mL/min was arbitrarily decided on the basis of the balance between the cooling effect and a clear vision of the surgical field during the bony work.

Efficacy and precision of the osteotomies, as well as the ability to spare the underlying soft tissues and neurovascular structures, were intraoperatively evaluated in all surgeries, by two different surgeons. 

In order to have a rough idea about the invasiveness of the piezoelectric cut, average blood loss was measured in all animals. The blood loss was calculated as the difference between the known amount of irrigation liquids attached to the peristaltic pump, and those present in the suction receptacles. An ANOVA test was performed between the three different classes of dogs and cats, namely large, medium-sized, and small ones. The *p*-value was set at <0.05.

However, because of the irrigation, this was assumed to be a broad evaluation of the real loss. 

Postoperatively, as a rule, all the animals operated on for a skull pathology underwent a CT scan. 

Regarding the spinal cases, an X-ray in both the projections was performed by default at a six-month follow-up, completed with a CT scan where the owner was available. Nevertheless, further imaging evaluations were decided on a case-by-case basis, according to the treated pathology.

The linearity and precision of osteotomies, and the incidence of osteosclerosis and bone resorption in the adjacent segments, were evaluated on imaging studies by two different surgeons.

The criteria adopted to objectively evaluate the long-term incidence of osteosclerosis and bone resorption at the level of the cut surfaces were derived by a modification of that reported by Bolm-Audorff and colleagues for the lumbar spine [37]. In particular, both on X-ray and CT scan, we considered as osteosclerotic at cut surfaces where the thickness of cortical bone extended into the cancellous bone by >2 mm.

A hematoxylin–eosin histological study of the bony specimens was carried out in all cases, to evaluate the presence of osteocytes, osteoblasts, and bone extra-cellular matrix at the osteotomized surface. In 17 cases of paraplegic or tetraplegic animals, where the owner asked for euthanasia, a postmortem harvesting of the spinal cord allowed us to evaluate the presence of mechanical or thermal damages to the nervous tissue and meninges.

## 3. Results

A total of 292 dogs and 32 cats were included in the study.

The number of large, medium-sized, and small dogs was 97, 121, and 74, respectively. In the cats’ group, 9, 19, and 4 were large, medium-sized, and small breeds. The average age of dogs was 6 ± 1.41, whereas mean age in cats was 6.6 ± 6. The male/female ratio was 2.8 for dogs and 0.7 for cats. 

The number of craniotomies and laminectomy/hemilaminectomy was 4 and 305, respectively. In the laminectomy/hemilaminectomy group, T11-L3 level was increased in 92% of cases. Fifteen were the ventral slots, and C5-C7 segment was involved in 94% of cases. Table 1 reports a complete spectrum of data about the neurosurgical procedures performed with the piezoelectric bone scalpel (Table 1).

In all surgeries, the edges of cuts were sharp and extremely precise. In several cases, the piezoelectric scalpel allowed us to achieve curved micrometric cuts that are otherwise difficult to be obtained by means of rotating instruments or saws. Moreover, in some spinal cases, the piezo scalpel was useful to perform the undercutting of the lateral aspect of the lamina without damaging the exposed dura. 

The operative field resulted in being free from blood and debris in all cases, thus facilitating a fast and safe recognition of all neighboring and underlying anatomical structures. Continuous irrigation and the thin and sharp profile of the different instruments’ tips and inserts allowed us to perform accurate and blood-free osteotomies, also in deep and difficult-to-access fields. No signs of overheating were observed on soft tissues or underlying neurovascular structures. Average blood loss was 51.6 mL in dogs and 25 mL in cats, respectively. However, no differences in blood loss were found between the large, medium, and small size in dogs and cats.

Table 2 and Table 3 report the average blood loss in dogs and cats, respectively.

Being the piezoelectric surgery based on the physical principle of the indirect piezo effect and cavitation, its efficacy was practically the same in dogs’ and cats’ laminectomies, regardless of the obvious differences in the thickness of the lamina. On the contrary, the length of the cutting tip may vary in dogs’ and cats’ groups.

A rapid learning curve about the use of the piezoelectric scalpel, characterized by a significant improvement of the confidence and dexterity with the instrument, was noted already after the firsts twenty treated cases. Table 4 reports the duration of the surgical procedures (Table 4).

In two trauma cases, in the dogs’ group, a mechanical damage attributable to the use of the piezoelectric bone scalpel was observed (0.6%). They consisted in a dural tear and an epidural hematoma.

Apart from an X-ray, a CT scan and MRI were performed postoperatively in 26 and 4 specific cases, respectively, where the animals underwent surgery because of a spinal traumatic or degenerative pathology.

The histologic study on bony specimens at the osteotomized surface showed the presence of live osteocytes and osteoblasts in 92% of cases (Figure 2). Harvested spinal cord in 17 postmortem cases allowed to appreciate the anatomical integrity of both the dura and the neural tissue (Figure 3).

### 3.1. Illustrative Cases 

#### 3.1.1. Case N° 1

Dog, male, nine years old, 40 kg, diagnosed with a lumbosacral stenosis on epidurography. He underwent a L7-S1 laminectomy by means of the piezoelectric bone scalpel. The procedure was fast and bloodless, and a wide and clear-cut decompressive laminectomy with a complete sparing of the underlying dura was performed (Figure 4). Two weeks after surgery, the walk significantly improved.

#### 3.1.2. Case N° 2

Dog, female, four years old, 0.8 kg, paraplegic, diagnosed with a T13-L1 fracture-luxation with hyperkyphosis secondary to a road accident. The dog underwent a thoracic laminectomy with a monolateral vertebral stabilization T13-L1 with plate and screws (Figure 5). By the six-month follow-up, the dog had partially recovered. 

#### 3.1.3. Case N° 3

Dog, female, crossbreed, six years old, 14 kg, who suffered from seizures due to a suspected left intracranial parietal meningioma. She underwent a lateral rostro tentorial craniotomy with the piezoelectric bone scalpel. No dura mater sinus was present. The scalpel was essential in sparing the underling dura and brain. The tumor was completely removed, and recovery was complete, without complications (Figure 6).

#### 3.1.4. Case N° 4

Dachshund, female, seven years old, 8 kg, suffering from a progressive tetraplegia due to a cervical disk extrusion involving the C2-C3 space. 

The dog underwent a ventral slot decompression by means of the piezoelectric bone scalpel with a curved long insert, which resulted in being very helpful, considering the depth of the surgical field compared to the dorsal approach (Figure 7). Two week later, the motor deficits significantly improved, and no signs of instability were documented.

#### 3.1.5. Case N° 5

Cat, male, three years old, 3 kg. Firearm trauma with a secondary spinal cord compression due to a retained lead shot. The cat was paraplegic. He underwent to a laminectomy and spinal cord decompression by means of the piezoelectric bone scalpel, where, once again, it was possible to spare completely the spinal cord (Figure 8). Recovery was very fast, without complications. 

## 4. Discussion

The results of the present study confirmed the reliability, effectiveness, safety, and minimal invasiveness of the piezoelectric osteotomy in veterinary neurosurgical procedures.

The physical principle of cavitation, at the base of the piezoelectric osteotomy, consists in the formation of vapor-filled bubbles generated by abrupt pressure changes of a liquid molecules. The subsequent collapse of these bubbles originates an energetic input which is transformed into a shock wave able to break the mineralized tissue [14,15,16,17,38,39].

Piezoelectric osteotomy differs itself from other techniques mainly for its micrometric and selective cut on the mineralized tissues, but also for allowing a heat-free and blood-free operating site. Piezoelectric cut is realized by mechanical vibrations in a range between 60 and 210 μm with a frequency of 29.000 times per second, thus obtaining an efficacy that, although slightly inferior to that of bone saws or burs, has been reported to not increase the overall duration of surgery [18,40,41,42].

The microscopic feature of the cut gives a high precision, which is paramount in several deep-seated surgical sites. About the selectivity, piezoelectric hand-piece produces microvibrations at a low frequency ultrasonic waves of 22–30 Hz. At this order of frequency, soft tissues and neurovascular structures are spared. 

About the power of the instrument, some presets are available, mainly varying according to the density of the tissue. As a rule, the greater the consistence of the tissue is, the greater the power required. The bone preset involved a power of 16 W and an average frequency of 30 KHz, as reported in the illustrative case 1.

Intra-operative irrigation is also subject to ultrasonic microvibrations, with these causing the breaking up of the liquid into very small particles. Liquid particles produce an excellent cooling of the bony surface, also making the operating site blood-free. Several studies have already confirmed the safety of piezosurgery [13,21,43,44,45].

Stelzle and Vercellotti reported that a key aspect related to the correct use of the instrument is the pressure load applied on the tip, with it largely affecting the overall efficacy and safety of the procedure. 

They and other authors reported that an excessive pressure load prevents microvibrations of the insert, and the total amount of energy not employed for cutting is converted into heat, ultimately leading to damage the soft tissues and neurovascular structures [13,18,40,46]. As a consequence, irrigation is essential for leading to the cavitation phenomenon and, at the same time, avoiding dangerous overheating. 

The cutting lines varied in shape, and, as reported in the Results, the piezoelectric scalpel allowed us to perform curved micrometric cuts, which were very useful in different surgical scenarios. About the inserts, the OT7 ones, characterized by an angled large saw tip, are recommended for coarse osteotomies. They are available in two different lengths, 10 and 20 mm, to be chosen on the basis of the depth of the bone. As a rule, the longest one is suitable for ventral slots, as reported in the illustrative Case 4, or also lumbosacral laminectomy. The tips have some landmarks on their lateral surface that act as depth meter. The thickness of the tips is 0.55 and 0.6 mm for the short and long one, respectively. In our experience, OT2 insert has been useful in high-precision osteotomies thanks to a 3 mm length sharp tip.

We found all the presets and working protocols to be suitable for most of the osteotomies.

The present study was conducted on dogs and cats affected by skull and spinal pathologies. This aspect allowed us to test the effectiveness, invasiveness, and safety of the instrument, especially in laminectomies and craniotomies. The precision and safety of the piezoelectric cut resulted in being particularly useful for hemilaminectomies, laminectomies, and craniotomies, where it was possible to protect the underlying dura, thus avoiding iatrogenic trauma. Nevertheless, a subperiosteal skeletonization is of the utmost importance in achieving the best performance of the scalpel in cutting the bone.

The two complications reported in the present series, namely a dural tear and an epidural hematoma, were both attributable to an accidental slippage of the instrument, with a consequent mechanical damage caused by the tip of the scalpel. Avoiding placing an excessive pressure on the instrument handpiece is important, also, to avoid these potential complications.

About the differences between dogs and cats, it should be stressed that we noted no differences in terms of efficacy of the cut. This aspect is attributable to the fact that piezoelectric surgery is based on the physical principles of indirect piezo effect and cavitation, which are both independent by the different thickness of the lamina in the two groups. 

All the aforementioned aspects make piezosurgery a very good tool for most of the complex skull and spinal neurosurgical approaches [47,48,49,50,51,52,53,54,55,56,57,58,59]. 

Conversely, a known disadvantage of the piezosurgery is that it is time-consuming [1,60,61]. However, in the authors’ experience, the slight increase in the overall operative time was largely counterbalanced by the safety of the technique. 

The minimal invasiveness of the piezo cut was also confirmed by the imaging follow-up, where the incidence of the osteosclerosis and bone resorption at the osteotomized surfaces was negligible (Figure 9a,b).

In the present study, whose major strength lies in the high number of treated cases, the non-aggressiveness and safety of the piezoelectric bone scalpel were also confirmed by the histology, which showed the absence of necrosis along with vitality of the osteocytes and osteoblasts. Similar results have been reported in the literature [62,63,64,65,66,67]. Piezosurgery has also been reported to interfere to a lesser extent than drills with the initial phases of the bone healing, mainly because it stimulates the bone morphogenetic proteins. A better control of iatrogenic inflammatory processes, along with the stimulation of bone remodeling has been also demonstrated [68,69,70,71,72,73].

The good outcomes reported in the present study allow us to consider the piezoelectric bone scalpel as a useful tool to be integrated into the neurosurgical armamentarium of the minimal invasive techniques [74,75,76,77,78].

## 5. Conclusions

The piezoelectric cut can be considered a minimal invasive osteotomy technique.

In our experience, it has proven to be precise, effective, and safe in veterinary craniotomies and laminectomies.

Its main strength lies in the ability to spare the soft tissues and neurovascular structures, being moreover free from the risk of thermal injury.

## Figures and Tables

**Figure 1 vetsci-07-00068-f001:**
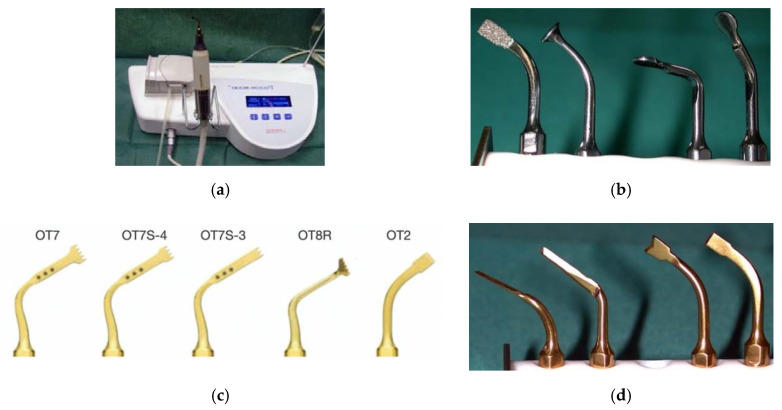
Mectron Piezosurgery^®^ bone scalpel. Power unit, hand-piece, and irrigation peristaltic module (**a**). Sets of tips (**b**–**d**).

**Figure 2 vetsci-07-00068-f002:**
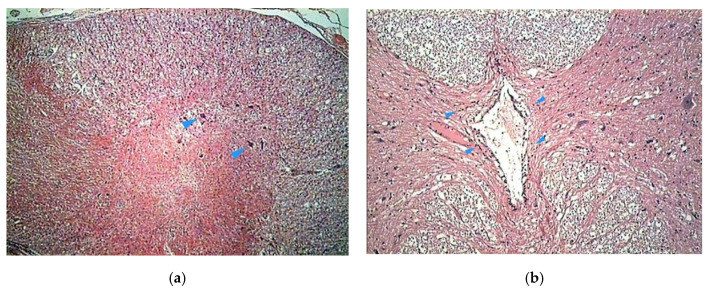
Histologic study executed on a dog’s bone specimen. Hematoxylin–eosin 2.5X (**a**) and 10X (**b**), showing the presence of live osteocytes and osteoblasts near the osteotomized surface (arrowheads).

**Figure 3 vetsci-07-00068-f003:**
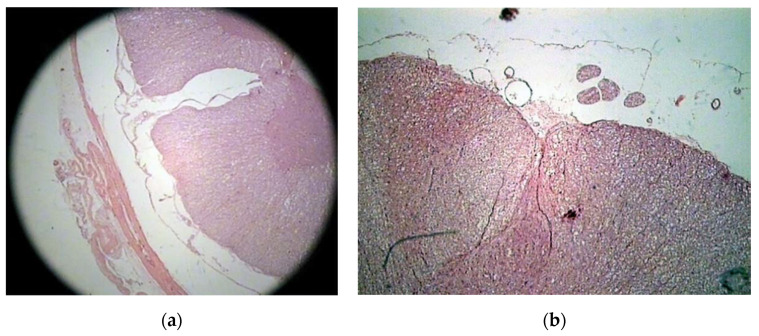
Histologic study executed on a cat’s spinal cord. Hematoxylin–eosin 2.5X, showing the perfect anatomical integrity of the spinal cord and dura after piezoelectric osteotomy (**a**,**b**).

**Figure 4 vetsci-07-00068-f004:**
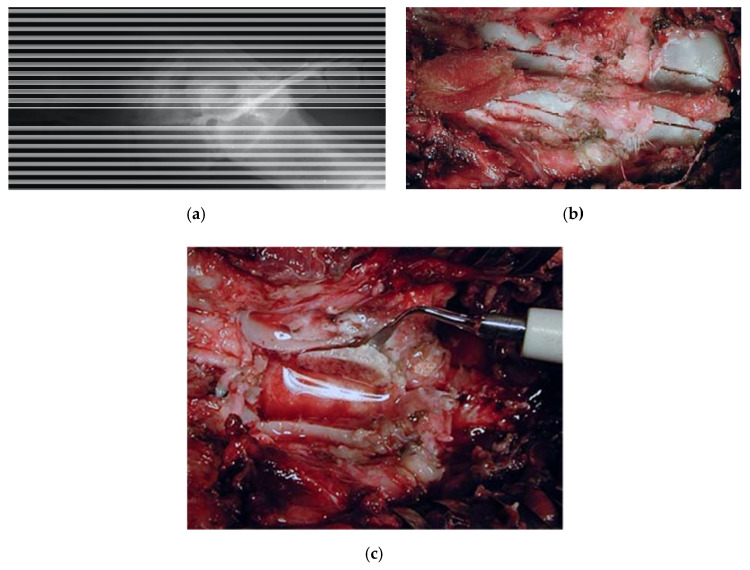
Dog, nine years old, diagnosed with a lumbosacral stenosis. (**a**) Epidurography showing a severe stenosis. (**b**) Skeletonization of the lumbosacral segment. Note the incision of the last lumbar and first sacral vertebra, along with the possibility of a further lateral extension (**c**) lumbar laminectomy. The thickness of the lamina measures 12 mm. Laminectomy was performed by means of the piezoelectric bone scalpel. An angled large saw tip (OT7, Osteotomy Tips Kit, Mectron Medical Technology, Genoa, Italy) was used. The bone preset involved a frequency of 30 KHz and a power of 16 W. A complete sparing of the underlying dura was observed after the osteotomy. OT7 designated tip also allowed to wide laterally the osteotomy in an easy way after and the dura exposure. Note that this tip allows cuts up to 15 mm of depth.

**Figure 5 vetsci-07-00068-f005:**
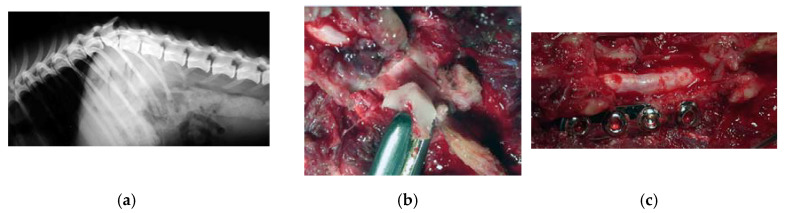
X-ray showing a T13-L1 fracture with hyperkyphosis in a paraplegic dog (**a**). A safe thoracic piezoelectric laminectomy was obtained with an angled sharp tip (OT2, Osteotomy Tips Kit, Mectron Medical Technology, Genoa, Italy) piezoelectric bone scalpel (**b**). Dorsal monolateral vertebral stabilization T13-L1 with plate and screws. Noteworthy, piezoelectric scalpel with a different straight tip was used as tapper for the screws placement (**c**).

**Figure 6 vetsci-07-00068-f006:**
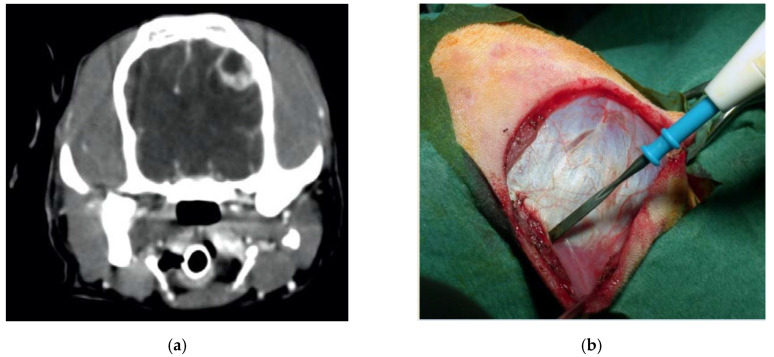
Preoperative contrast-enhanced CT scan, showing a suspected left parietal meningioma (**a**). Intraoperative pictures showing the initial steps of the craniotomy (**b**). The bone flap was cut with piezoelectric scalpel and an angled large saw tip (OT7, Osteotomy Tips Kit, Mectron Medical Technology, Genoa, Italy) (**c**,**d**). The dura was completely preserved (**e**). En bloc removal of meningioma (**f**). Postoperative CT scan performed at six-month follow-up, showing the sharpness of the edges of the craniotomy. No osteosclerosis or bony resorption was evident (**g**).

**Figure 7 vetsci-07-00068-f007:**
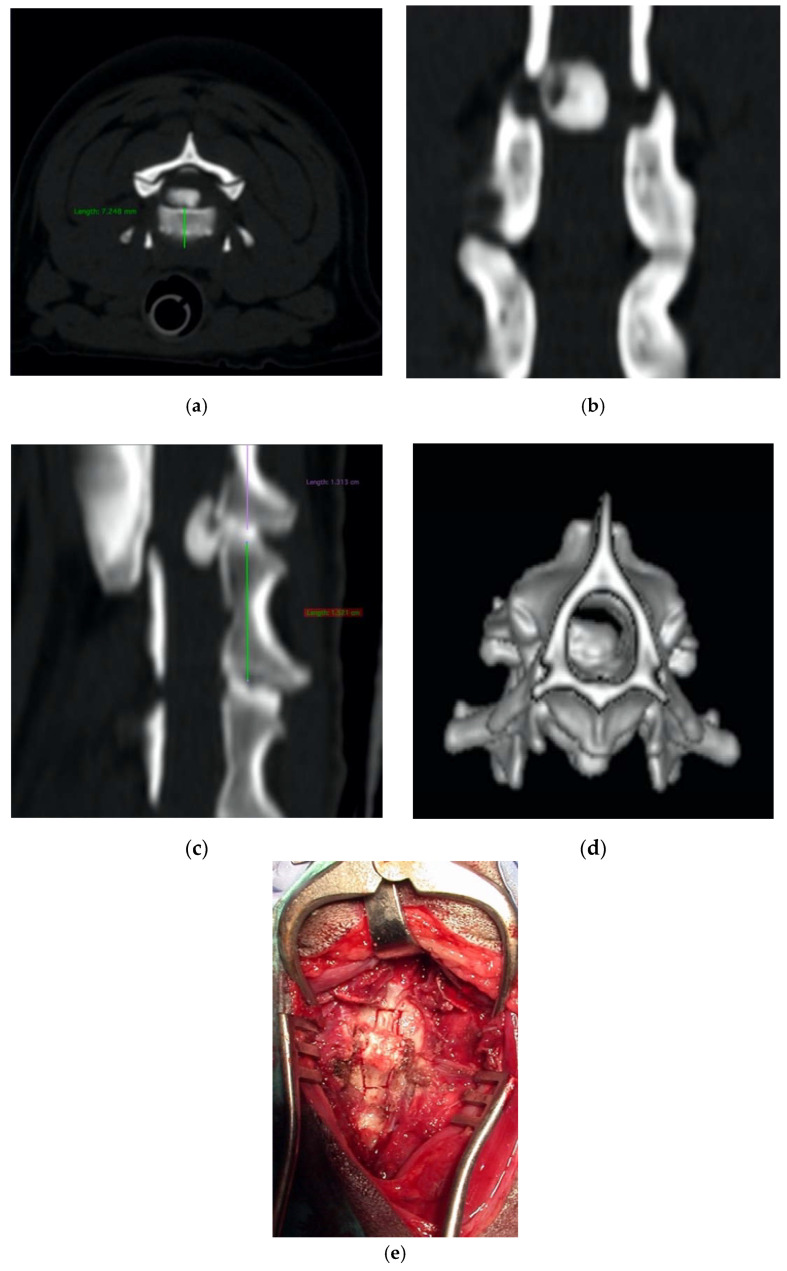
Axial (**a**), coronal (**b**), sagittal (**c**), and 3D (**d**) cervical CT scan showing a C2-C3 disk extrusion in a seven-year-old dachshund suffering from a progressive tetraplegia. Ventral slot decompression was obtained by means of long angled large saw tip (OT7, Osteotomy Tips Kit, Mectron Medical Technology, Genoa, Italy) (**e**).

**Figure 8 vetsci-07-00068-f008:**
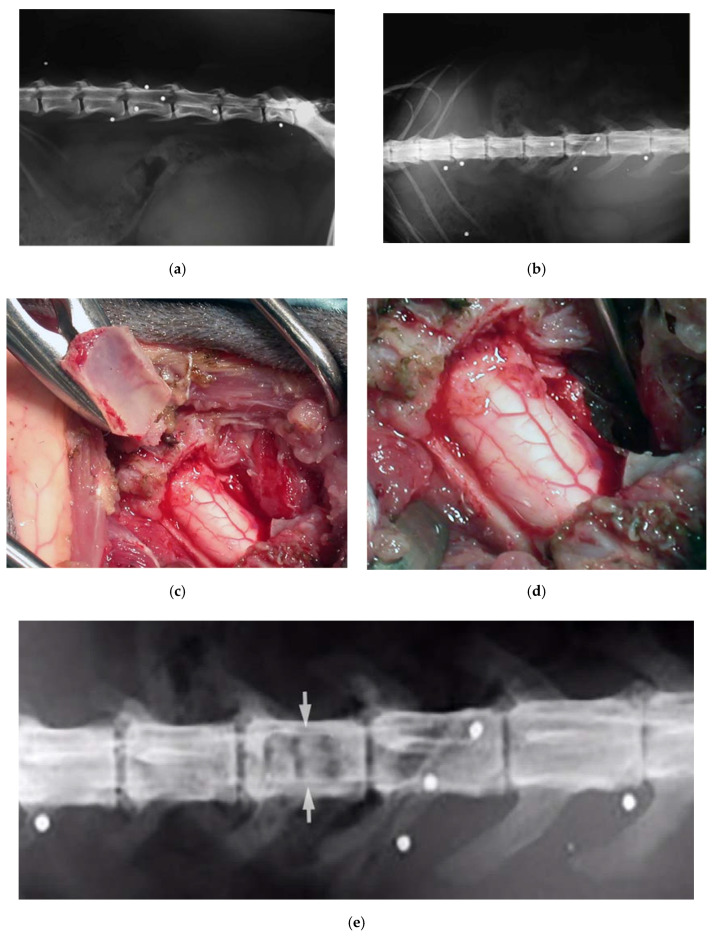
X-ray in latero-lateral (**a**) and ventro-dorsal (**b**) projection, showing multiple retained lead shots in a cat paraplegic because victim of a firearm trauma. One of the intrathecal lead shots caused a spinal cord contusion and partial compression. A subdural hematoma was also present. Intraoperative pictures showing lumbar spinal cord after decompressive one-level laminectomy performed with piezoelectric scalpel and an angled small saw tip (OT7S-3, Osteotomy Tips Kit, Mectron Medical Technology, Genoa, Italy) (**c**,**d**). Six-months postoperative ventro-dorsal X-ray showing the site of the laminectomy (**e**).

**Figure 9 vetsci-07-00068-f009:**
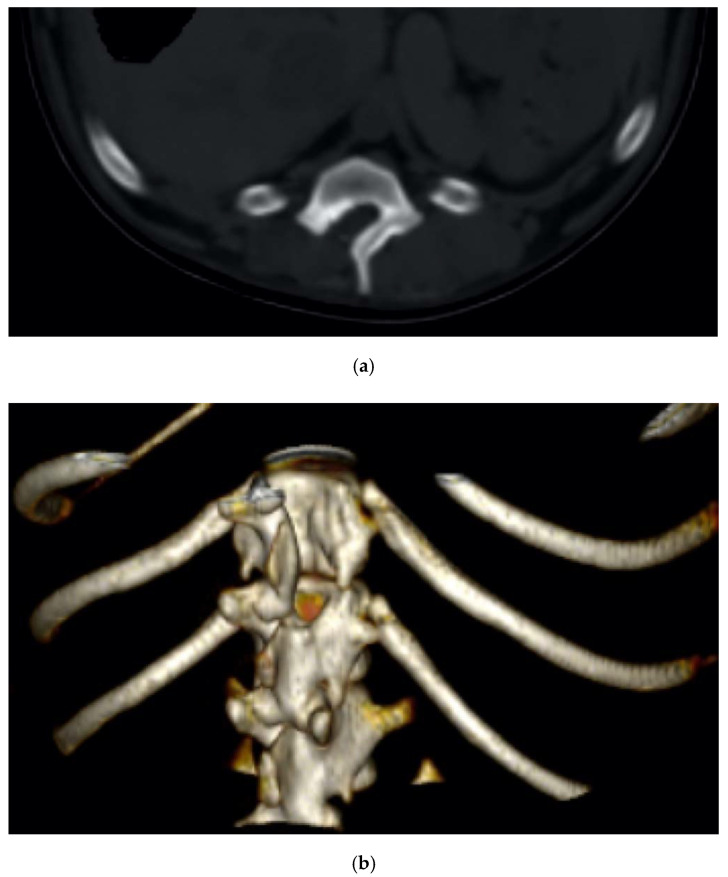
Two-dimensional (**a**) and 3D (**b**) CT scan of an explicative case of dorsal hemilaminectomy performed in a dachshund diagnosed with a spinal cord compression.

**Table 1 vetsci-07-00068-t001:** Data about neurosurgical procedures performed with the piezoelectric bone scalpel.

Site			Patient				Total
			Dogs (N° 292)			Cats (N° 32)	
			Field of Pathology				
			Traumatic	Degenerative	Neoplastic	Traumatic	
Site	Skull		0	0	4 (extra-axial)	0	4
	Spine	Cervical	0	12	2	0	12
		Thoracic	24	13	3	0	40
		Thoracolumbar	37	4	6	24	73
		Lumbar	14	62	2	0	78
		Lumbosacral	74	34	1	8	117
Surgeries (N°)			149	125	18	32	324
Average blood loss (mL)			52	47	56	25	180
Complications (N°)			2	0	1	0	3

**Table 2 vetsci-07-00068-t002:** Data about average blood loss according to dogs’ size.

Size	Dogs (N°)	Weight (lbs)		Blood Loss (mL)		
		Average	Sd	Average	Sd	ANOVA
Large (≥50 lbs)	97	104.3	34.3	51.0	17.7	0.32
Medium (21–50 lbs)	121	35.2	8.9	53.3	16.8	
Small (1–20 lbs)	74	10.4	6.1	49.8	16.4	

**Table 3 vetsci-07-00068-t003:** Data about average blood loss according to cats’ size.

Breed	Cats (N°)	Weight (lbs)		Blood Loss (mL)		
		Average	Sd	Average	Sd	ANOVA
Large (≥14 lbs)	9	15.3	1.1	25	6.3	0.61
Medium (9–13 lbs)	19	11.5	1.2	26.4	6.4	
Small (1–8 lbs)	4	6.5	1.9	6.5	1.9	

**Table 4 vetsci-07-00068-t004:** Duration of surgical procedures.

Site	Duration of Surgery (Average min)	
	Dogs	Cats
Skull	72	67
Cranial Cervical Spine	60	56
Caudal Cervical Spine	48	43
Thoracic	20	16
Thoracolumbar	22	17
Lumbosacral	17	14
Lumbar	26	23

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
