# Peer review of "Minimal Invasive Piezoelectric Osteotomy in Neurosurgery: Technic, Applications, and Clinical Outcomes of a Retrospective Case Series"

_vetsci, 2020, doi:10.3390/vetsci7020068_

Round 1
Author Response
Minimal Invasive Piezoelectric Osteotomy in Neurosurgery: Technic, Applications, and Clinical Outcomes of a Retrospective Case Series
Response to Reviewer 1
We want to thank the Reviewer# 1 for his comments and suggestions that have been very precious for us in order to improve the quality and clarity of our manuscript.
Below, we report an itemized, point-by-point response to the Reviewers’ kind remarks.
All the changes in the manuscript have been reported in track change mode ON.
Reviewer #1
- “The topic of the paper is interesting and the number of cases treated is relevant. In fact this topic is hardly discussed in veterinary medicine, as instead it has already been treated in human medicine, especially in odontostomatological surgery.”
We thank the Reviewer and completely agree with potential interest of the topic in veterinary medicine.
- “In my opinion, because of the potential it has in veterinary medicine, the paper must be rewritten by better emphasizing the characteristics of osteotomy, ease of use, safety on soft tissues and on the spinal cord in the piezoelectric surgery. Furthermore, patients-selection and groups-selection should be done in a better way perhaps by considering the types of surgery performed.”
Thank you again for this precious suggestion, based on which we have rewritten our manuscript highlighting and stressing some technical aspects of the piezoelectric osteotomy, along with its potential and non-negligible advantages in clinical practice. Furthermore, we have differentiated the characteristics of the craniotomy and spinal osteotomies (laminectomy, hemilaminectomy and ventral slot) in a better way in order to emphasizes the benefits and the limitations (see below).
- “All of this should be done in order to give value to the number of cases analysed, otherwise there is a risk that the whole research will not be considered valuable, which is something not wanted.”
Thank you for these valuable advices that we have seriously taken into account to improve the quality of our article.
- “As the research has potential, the number of cases must be underlined, in order to give more scientific value to the paper.”
About this point, we have stressed number of treated cases to increase the scientific value of our study.
- “Moreover, I believe that the number clinical cases showed could have been reduced and presented in a way that would have highlighted the effects of Piezosurgery.
We have removed the illustrative case 5, 6 and 7, better elucidating in the remaining ones the potential benefit of the piezosurgery.
- “I would add long-term CT images to show the bone healing”.
We fully agree also with this point. Accordingly, we have added some long-term CT (Fig. 6 g; Fig. 9 a and b) and X-ray image (Fig. 8 e) with the aim to show the bone healing intended as the reparative process of the bone at the cut surfaces.
Introduction
- “I would suggest to the authors to explain in more detail the principles of piezoelectric with related micromechanical, thermal, chemical and cavitation physical effects on the tissues. Only after these considerations explain the piezosurgery instrument with its indications.”
We are grateful to the kind Reviewer for this point. We totally agree with the need to better explain the physical aspects of the cavitation phenomenon. Accordingly, we have added the following sentence: “Piezoelectric osteotomy is based upon the physical principle of cavitation. Cavitation phenomenon involves the formation of vapor-filled cavities secondary to the abrupt and rapid changes of pressure in a liquid. The input coming from a high-pressure source causes collapse of these cavities and generate a shock wave which propagates in the tissue. In turn, the shock wave causes cell detachment from substrate, significant changes in focal adhesion and biomechanics, ultimately inducing a mechanical cut on the mineralized tissue.
In piezoelectric surgery, the changes of pressure are generated by a spatial deformation of the piezoelectric crystals when they are subject to an electric field having a given ultrasonic frequency. This mechanism is also called indirect piezo effect. The resulting oscillations are then amplified and transferred onto an instrument tip.”.
- Line 48: “I would remove oral surgery because it is not relevant in this paper.”
We have avoided to refer to oral surgery because, as rightly suggested, it is not relevant.
- Line 50: “When the authors write ‘heat-free’ surgery I would also explain the physical effect of cavitation. Cavitation effects can increase the temperature of the tissue but at the same time promotes the absence of blood.”
Thanks to this valuable suggestion, we have better explained the concept as it follows: “This piezoelectric osteotomy is also considered as a “heat-free” technique.
In fact, although the cavitation effects can increase the temperature of the neighboring tissues, the dissipation of the heat coming from the concomitant irrigation, practically leads to avoid any thermal injury. It also promotes the absence of blood and debris in the surgical field”.
Materials and Methods
- Line 68: “Define the criteria for creating the neurosurgery and orthopedics groups. Explain which surgical procedures characterized the two groups.”
We thank you very much the Reviewer for this important point. On the bases of the suggestion of this and other Reviewers we have better classified the procedures, intending all of them as neurosurgical rather than orthopedic. Accordingly, we have changed the title of the article as it follows: “Minimal Invasive Piezoelectric Osteotomy in Neurosurgery: Technic, Applications, and Clinical Outcomes of a Retrospective Case Series”, and also better clarified the inclusion and exclusion criteria of the study as it follows: “Inclusion criteria involved dogs and cats which required a neurosurgical procedure comprehending a spinal or skull osteotomy because of a traumatic, degenerative or neoplastic lesion. Animals older than 10 years-old or harboring more than a single neurosurgical pathology were excluded.”. Furthermore, we have limited to the only neurosurgical cases the results.
- Line 70: “Explain when it was decided to do X-ray or CT and why. Explain also what you want to see with X-ray and what you want to see with CT”.
We completely agree with the need for better explain when it was decided to do X-ray or CT. Therefore, we have clarified this point as it follows: “In all spinal cases, the patients underwent to a ventro-dorsal and latero-lateral projection X-ray. A spinal computed tomography (CT) scan and magnetic resonance imaging (MRI) were proposed in all cases but performed only in those where the owner was available. Conversely, CT scan was instead mandatory in all cranial cases. Images were reviewed and edited on a DICOM imaging workstation (Osirix DICOM Viewer@, Pixmeo, Bernex, Switzerland) which also allowed some measurements.”
In regard to the explanation of what we wanted to see, we have added the following sentence: “In spinal cases, X-ray and CT scan were both aimed to study the bony component of the pathology. 3D rendering of the spinal CT scan was also performed in all cases to study in more detail the site and cause of compression. In cranial cases, 3D CT scan allowed to better localize the lesion. MRI was paramount to study the neurovascular anatomical structures and soft tissues..”
- Line 75: “Define better the case studies: how many large dogs, medium-sized dogs and small dogs. This is very important to understand the dimensions\thickness of the bone segments. Define the size of the cats. Define age, sex of animals, etc.. Define the type of surgery performed: craniotomy, laminectomy, hemilaminectomy, ventral slot and specify at what level of the spine it is done. Moreover, it is not clear what type of osteotomy the subjects of the orthopedic group performed.”
We thank once again the kind Reviewer. We have moved these data in Results and specified all the required details as it follows: “A total of 292 dogs and 32 cats were included in the study. The number of large, medium-sized and small dogs was 97, 121 and 74, respectively. In the cats’ group, 9, 19 and 4 were large, medium-sized and small breed. Average age of dogs was 6 ± 1.41, whereas mean age in cats was 6.6 ± 6. Male/female ratio was 2.8 for dogs and 0.7 for cats. The number of craniotomies and laminectomy/hemilaminectomy was 4 and 305, respectively. In the laminectomy/hemilaminectomy group, T11 -L3 level was interested in 92% of cases. Fifteen were the ventral slots, and C5-C7 segment was involved in 94% of cases”.
We also want to stress that, after a detailed reassessment about the type of procedures, all of them were considered as neurosurgical.
- Line 86: “Explain better what the two surgeons evaluated in order to make a judgment on the cutting of the Piezosurgery.”
About this point, we have clarified in Materials and Methods what were the criteria used to objectively evaluate the cutting of the Piezosurgery: “The criteria adopted to objectively evaluate the long-term incidence of osteosclerosis and bone resorption at the level of the cut surfaces were derived by a modification of that reported by Bolm-Audorff and colleagues for the lumbar spine. In particular, both on X-ray and CT scan, we considered as osteosclerotic a cut surfaces where the thickness of cortical bone extended into the cancellous bone by > 2 mm.”.
- Line 87: “Explain why blood loss has been measured, how this measurement was performed, and how important it is. The Piezosurgery has an irrigation pump. This aspect can be influenced by the duration of the surgery and by the thickness of the bone to be cut.”
We have clarified this aspect in Materials and Methods by means of the following sentence: “In order to have a rough idea about the invasiveness of the piezoelectric cut, average blood loss was measured in all animals. The blood loss was calculated as the difference between the known amount of irrigation liquids attached to the peristaltic pump, and those present in the suction receptals.”.
- Line 87: “both groups”: “does it mean “the group of dogs and cats” or “the neurosurgery and orthopedics groups”? It is not clear why they have not been clearly defined before.”
The misleading term “both groups” has been changed with “all animals” as reported above.
- Line 88: “X-rays were taken in the post-op: for what purpose and in which animals has it been chosen to perform CT and/or MRI?. In my opinion, it is difficult to evaluate the linearity of the cut with X-rays. Furthermore osteosclerosis and bone resorption are long-term complications. CT is the best imaging exam while MRI is not indicated to investigate bone. In lines 69-70 the MRI was not mentioned.”
We completely agree with the Reviewer on all these observations, especially with the difficult to evaluate the linearity of the cut with X-rays. In fact, as reported above, we used the radiological criteria reported by Bolm-Audorff and colleagues to assess the incidence of osteosclerosis and bone resorption, which in turn, was assumed to be an indirect estimation of the linearity of the piezo cut. Therefore, we have specified in Materials and Methods this concept: “The criteria adopted to objectively evaluate the long-term incidence of osteosclerosis and bone resorption at the level of the cut surfaces were derived by a modification of that reported by Bolm-Audorff and colleagues for the lumbar spine. In particular, both on X-ray and CT scan, we considered as osteosclerotic a cut surfaces where the thickness of cortical bone extended into the cancellous bone by > 2 mm.”.
Furthermore, we have clarified that: “In all spinal cases, the patients underwent to a ventro-dorsal and latero-lateral projection X-ray. A spinal computed tomography (CT) scan and magnetic resonance imaging (MRI) were proposed in all cases but performed only in those where the owner was available. Conversely, CT scan was instead mandatory in all cranial cases. Images were reviewed and edited on a DICOM imaging workstation (Osirix DICOM Viewer@, Pixmeo, Bernex, Switzerland) which also allowed some measurements.
In spinal cases, X-ray and CT scan were both aimed to study the bony component of the pathology. 3D rendering of the spinal CT scan was also performed in all cases to study in more detail the site and cause of compression. In cranial cases, 3D CT scan allowed to better localize the lesion. MRI was paramount to study the neurovascular anatomical structures and soft tissues.”, thus explaining which were our indication for the execution of CT scan and MRI.
- Line 93: “The period mark is missing at the end of the sentence.”
We have corrected this typo error.
- Line 94: “How many and which were the “selected cases” and which surgery have they done?”
We have moved this data in Results, and we have specified that: “Apart from X-ray, CT scan and MRI were performed postoperatively in 26 and 4 specific cases, respectively, where the animals underwent to surgery because of a spinal traumatic or degenerative pathology.”.
- Line 96: What was evaluated in the follow-up and what was evaluated before the surgery?
We have clarified what was preoperatively evaluated: “In spinal cases, X-ray and CT scan were both aimed to study the bony component of the pathology. 3D rendering of the spinal CT scan was also performed in all cases to study in more detail the site and cause of compression. In cranial cases, 3D CT scan allowed to better localize the lesion. MRI was paramount to study the neurovascular anatomical structures and soft tissues.”.
Furthermore, we have specified which were the parameters assessed during the follow-up: “The criteria adopted to objectively evaluate the long-term incidence of osteosclerosis and bone resorption at the level of the cut surfaces were derived by a modification of that reported by Bolm-Audorff and colleagues for the lumbar spine. In particular, both on X-ray and CT scan, we considered as osteosclerotic a cut surfaces where the thickness of cortical bone extended into the cancellous bone by > 2 mm.”.
- Line 100: in Tab. 1 it is not clear which type of orthopedic surgery has been performed: it seems that only neurosurgery has been performed. What is meant by “traumatic”? What by “degenerative”? Which diseases do you refer to? Does “Neoplastic” refer to bone or to intracranial neoplasms and\or within the vertebral canal?
Replace in table 1 “cranial” with “skull”.
We thank the Reviewer also for this point. Once again, we would like to clarify that we have now classified our procedures all as neurosurgical. Moreover, on a par with what reported in several studies, we intended as traumatic field of pathology basically the bone fractures, whereas disk protrusions, stenosis, Wobbler syndrome etc. were considered as degenerative conditions. We have replaced the term “cranial” with the term “skull” and, about the skull, we have specified in Table 1 that all the 4 cases involved extra-axial lesions, namely meningiomas.
Results
- Line 103-105: The curved cut can also be done with a small burr. Explain better and underline this aspect in the discussion.
We totally agree with the Reviewer about the possibility to perform curved cut also with a small burr. Nevertheless, as known, these are generally micrometric, being instead the curved cuts allowed by the piezo scalpel micrometric and extremely precise. These have been the results of our experience. Furthermore, is deep surgical field, piezoelectric scalpel permitted to perform the undercutting of the lateral aspects of the lamina even after the dura exposure. In these circumstances, the dura was almost never damaged by the instrument. We have better explained these concepts as it follows: “In all surgeries the edges of cuts were sharp and extremely precise. In several cases, the piezoelectric scalpel allowed to achieve curved micrometric cuts otherwise difficult to be obtained by means of rotating instruments or saws. Moreover, in some spinal cases, the piezo scalpel was useful to perform the undercutting of the lateral aspect of the lamina without damaging the exposed dura.”.
- Line 108: “Rapid learning curve”: how many cases should be done? Try to quantify this data.
Thank you also for this good question. In our experience, we noted a significant improvement of the confidence and dexterity with the instrument, already after the firsts twenty treated cases. We have reported these further data coming from our experience: “A rapid learning curve about the use of the piezoelectric scalpel, characterized by a significant improvement of the confidence and dexterity with the instrument, was noted already after the firsts twenty treated cases.”.
- Line 110: “It is not clear if the trauma was given by the pre-op injury or by the action of the Piezosurgery. Explain better.”
We have better explained that the mechanical injury we talk about has been referred to the action of the Piezosurgery: “In two trauma cases, in the dogs’ group, a mechanical damage attributable to the use of the piezoelectric bone scalpel was observed (0.6%).”.
- Line 112: “As you measured the blood loss (m & m), here it is highlighted that during the osteotomy there is always continuous irrigation. In order to obtain these results a better explanation of the method (in m & m) is needed.”
Thank you also for this point. We totally agree about the fact that the continuous irrigation certainly biases the interpretation of these data. However, while admitting this overestimation, we measured the amount of blood lost intraoperatively in order to have a general idea and an initial impression of the invasiveness of the instruments on the soft tissues. Accordingly, we have stressed all the limits coming from this data and better explained the method as it follows:“In order to have a rough idea about the invasiveness of the piezoelectric cut, average blood loss was measured in all animals. The blood loss was calculated as the difference between the known amount of irrigation liquids attached to the peristaltic pump, and those present in the suction receptals.”.
- Line 113: “In my opinion with X-rays it is very difficult to affirm this, CT gives better information.”
We obviously agree with the kind Reviewer also on this point and, as a consequence, we have better explained in Materials and Methods that, depending by the availability of the owner, we preferred to perform CT scan in every possible cases: “A spinal computed tomography (CT) scan and magnetic resonance imaging (MRI) were proposed in all cases but performed only in those where the owner was available.”
- Line 115: “In which and how many cases was the spinal cord collected? What kind of surgery have they undergone? what was the cause of death?”
We have clarified this point as it follows: “In 17 paraplegic or tetraplegic animals, where the owner asked for euthanasia, a post-mortem harvesting of the spinal cord allowed to evaluate the presence of mechanical or thermal damages to the nervous tissue and meninges.”.
- “In this section I would also write the differences that have been recorded between dogs and cats. Bone thickness and vertebral canal are certainly different.”
Based in this further suggestion, we have clarified the difference we found in dogs’ and cats’ group, which basically consisted in the different length of the cutting tips employed. No differences were instead observed in terms of efficacy of the cut. In Results we have added the following sentence: “Being the piezoelectric surgery based on the physical principle of the indirect piezo effect and cavitation, its efficacy was practically the same in dogs’ and cats’ laminectomies, regardless by the obvious differences in the thickness of the lamina. On the contrary, the length of the cutting tip may vary in dogs’ and cats’ group.”.
CASE 1
- Line 123: “Orient the images in the same direction. indicate which blade and frequency were used and how thick the bone lamina was. The paper must give indications on the use of the Piezosurgery. In fig. 4 a I would write epidurography. In fig. 4 d you don’t see the cauda well, do you have a better picture?”
We have oriented the images in the same direction. The type of blade, frequency and power used have been specified. The thickness of the lamina has been reported, as well as the term “epidurography” in Fig. 4 a. Unfortunately, we don’t have a better picture, but we have magnified the Fig. 4 d to better show the cauda equina.
CASE 2
- Line 132: “Replace “She” with “The dog”.
We have replaced “she” with “The dog”.
- Line 133: Replace “postero lateral internal arthrodesis” with “monolateral vertebral stabilization T13-L1 with plate and screws”. The term postero-lateral is a terminology used in human medicine.
Based on this suggestion, we have changed the sentence as it follows: “The dog underwent to a thoracic laminectomy with a dorsal monolateral vertebral stabilization T13-L1 with plate and screws (Fig. 5).”.
- Line 135: “At the end of the sentence there are two period marks”.
We have corrected this typo error.
CASE 3
- Linea 137: “I would replace “an intracranial convexity meningioma” with “a suspected intracranial meningioma”.
We have replaced “an intracranial convexity meningioma” with “a suspected intracranial parietal meningioma”.
- Linea 138: “Remove “convexity” and write only “craniotomy”. Explain the localization of the mass and the type of access to the skull. Also explain the presence or absence of the dura mater sinus.”
We have removed “convexity”. We have specified that the meningioma affected the parietal area. We have also reported the type of access, and clarified that no dura mater sinus was present.
All these details have been summarized in the following sentences: “Dog, female, cross breed, 6 years-old, 14 kg, who suffered from seizures due to a suspected left intracranial parietal meningioma. He underwent to a lateral rostro tentorial craniotomy with the piezoelectric bone scalpel. No dura mater sinus was present.”.
- Line 142: “Remove “convexity” and write “intracranial meningioma”
We have removed “convexity”.
- Linea 143: “Remove “En bloc removal of meningioma (g-h)”. g-h pictures are not necessary to demonstrate osteotomy result”.
We thank you also for this point. We have removed the unnecessary Fig. h. Nevertheless, another Reviewer suggested to maintain the Fig. g, along with its figure caption, in order to document the type of lesion.
CASE 4
- “Specify the localization (in order to understand the image 7a), the depth and thickness of the vertebral body, the type of insert and the power used. this information should be considered in Discussion. Discuss also the vertebral instability it might generate. Here the ventral slot seems to me rather wide. explain if there were problems with the venous sinus. It would be useful to also show a picture of the type of insert used.”
Thank also for these important observations, according to which we have better defined the characteristics of the treated case as it follows: “Dachshund, female, 7 years-old, 8 kg, suffering from a progressive tetraplegia due to a cervical disk extrusion involving the C3-C4 space.
The dog underwent to a ventral slot decompression by means of the piezoelectric bone scalpel with a curved long insert which resulted very helpful considering the depth of the surgical field compared to the dorsal approach (Fig. 7). Two week later, the motor deficits significantly improved, and no signs of instability were documented.”.
Furthermore, no problems with the venous sinus occurred in this case.
In addition, we have specified the type of instrument used in the Figure Legend section: “Figure 7. Axial (a), coronal (b), sagittal (c) and 3D (d) cervical CT scan showing a C3-C4 disk extrusion in a 7 years-old dachshund suffering from a progressive tetraplegia. Ventral slot decompression was obtained by means of long angled large saw tip (OT7, Osteotomy Tips Kit, Mectron Medical Technology, Genoa, Italy) (e).”
- Line 145: “Replace “quadriplegia” with “tetraplegia”
- Line 152: “Replace “quadriplegia” with “tetraplegia”
In both the points we have replaced “quadriplegia” with “tetraplegia”.
CASE 5
- “Indicate localization of osteotomy and neurological deficits of the case. numbness is a subjective symptom that can be reported by a human and not by an animal. the right term is paraparesis, paraplegia, proprioception, etc.
- Line 156: “posterior hemilaminectomy” is a term used in human medicine. use “right” or “left hemilaminectomy” and specify the localization.
- Line 158: Remove “nerves”. When a hemilaminectomy is performed, the spinal cord is mainly exposed and the nerve root can be exposed.
- 8 a: The picture is unclear and does not show compression of the spinal cord well.
- Line 159: Latero-lateral X-ray (myelography). Explain the localization.
- Line 160: Specify the various types of inserts used.”
Based on the kind suggestions of the Reviewer (“Moreover, I believe that the number clinical cases showed could have been reduced, and presented in a way that would have highlighted the effects of Piezosurgery), we have removed the illustrative case 5.
CASE 6
- “Line 65: Cervical X-ray (myelography). there is no picture of the column in severe flexion.
Change the word “anterior” (human medicine) with “ventral”. - 9 a-b: The compression is in the C5-C6 space. Fig. 9 c: the compression is in the C4-C5 space.
- 9 d-e: The compressions are not clear: specify the location and indicate with arrows. However, all myelography pictures are not needed. The paper must make a contribution on osteotomy.
- It is better to use surgical pictures and highlight the cut of the Piezosurgery, the tIickness of the lamina, the type of insert and the power used.”
Based on the kind suggestions of the Reviewer (“Moreover, I believe that the number clinical cases showed could have been reduced, and presented in a way that would have highlighted the effects of Piezosurgery), we have removed the illustrative case 6.
CASE 7
- Line 169: “Change “dorso-lumbar laminectomy” with “laminectomy T13-L1”.
- Line 172: “Remove “dorso-lumbar, traumatic fracture of vetebral body of T13” and repalce with “traumatic fracture of the body of T13 with axial longitudinal deviation of the column” (sagittal mielogram is missing). Also here it is better to show the egg ostectomy, the cutting surface and explain the power of the piezosurgery used. (This case is a cat).”.
Based on the kind suggestions of the Reviewer (“Moreover, I believe that the number clinical cases showed could have been reduced, and presented in a way that would have highlighted the effects of Piezosurgery), we have removed the illustrative case 7.
CASE 8
- Line 175: “Indicate the localization of the lesion and neurological signs. specify whether it is a compression or a contusion/concussion of the spinal cord.”.
Thank you for this point. In the description of the case and the figure legends we have better clarified these data.
“Cat, male, 3 years-old, 3 kg. Firearm trauma with a secondary spinal cord compression due to a retained lead shot. The cat was paraplegic. He underwent to a laminectomy and spinal cord decompression by means of the piezoelectric bone scalpel where, one again, it was possible to spare completely the spinal cord (Fig. 8). Recovery was very fast without complications.”.
“Figure 8. X-ray in latero-lateral (a) and ventro-dorsal (b) projection showing multiple retained lead shots in a cat paraplegic because victim of a firearm trauma. One of the intrathecal lead shot caused a spinal cord contusion and partial compression. A subdural hematoma was also present. Intraoperative pictures showing lumbar spinal cord after decompressive one-level laminectomy performed with piezoelectric scalpel and an angled small saw tip (OT7S-3, Osteotomy Tips Kit, Mectron Medical Technology, Genoa, Italy) (c-d). Six-months post-operative ventro-dorsal X-ray showing the site of the laminectomy (e).”.
- 11: “If the fragment is paravertebral it cannot be a compression. the images show three shotgun bullets on L4. only the most caudal bullet appears to be within the vertebral canal. Later we talk about thoracic spinal cord ????, also subdural hematoma is noted.”.
We are grateful to the Reviewer for these observations. We have, in fact, rewritten the figure legend as it follows: “Figure 8. X-ray in latero-lateral (a) and ventro-dorsal (b) projection showing multiple retained lead shots in a cat paraplegic because victim of a firearm trauma. One of the intrathecal lead shot caused a spinal cord contusion and partial compression. A subdural hematoma was also present. Intraoperative pictures showing lumbar spinal cord after decompressive one-level laminectomy performed with piezoelectric scalpel and an angled small saw tip (OT7S-3, Osteotomy Tips Kit, Mectron Medical Technology, Genoa, Italy) (c-d). Six-months post-operative ventro-dorsal X-ray showing the site of the laminectomy (e).”.
We regret for having reported “thoracic” rather than “lumbar” spinal cord, but it was a typo mistake.
- In this part of the results the evaluation of surgeons in the various groups and the evaluation of the follow up are missing. I suggest making a table with objective assessments.
On the basis of this suggestion, we have added the following sentences regarding the evaluation of surgeons and the follow-up time for the groups in Materials and Methods section: “Postoperatively, as a rule, all the animal operated for a skull pathology, underwent to CT scan. Regarding the spinal cases, an X-ray in both the projections was performed by default at six-month follow-up, completed with a CT scan where the owner was available. Nevertheless, further imaging evaluations were decided on a case-by-case basis according to the treated pathology. The linearity and precision of osteotomies, and the incidence of osteosclerosis and bone resorption in the adjacent segments, were evaluated on imaging studies by two different surgeons.
The criteria adopted to objectively evaluate the long-term incidence of osteosclerosis and bone resorption at the level of the cut surfaces were derived by a modification of that reported by Bolm-Audorff and colleagues for the lumbar spine. In particular, both on X-ray and CT scan, we considered as osteosclerotic a cut surfaces where the thickness of cortical bone extended into the cancellous bone by > 2 mm.”; and in the Results: “Apart from X-ray, CT scan and MRI were performed postoperatively in 26 and 4 specific cases, respectively, where the animals underwent to surgery because of a spinal traumatic or degenerative pathology.”
Discussion
In the discussion, in addition to what has been written, I would discuss:
- “the phenomenon of cavitation”
In order to discuss the phenomenon of cavitation, we have added the following sentence: “The physical principle of cavitation, at the base of the piezoelectric osteotomy, consists in the formation of vapor-filled bubbles generated by abrupt pressure changes of a liquid molecules. The subsequent collapse of these bubbles originates an energetic input which is transformed in a shock wave able to break the mineralized tissue.”
- “the differences between the two groups and between dog and cat”
As clarified before, we have better classified all the procedures performed as neurosurgical. Concerning instead the differences about dogs and cats, we have added in discussion the following sentence: “About the differences between dogs and cats, it should be stressed that we noted no differences in terms of efficacy of the cut. This aspect is attributable to the fact that piezoelectric surgery is based on the physical principles of indirect piezo effect and cavitation, which are both independent by the different thickness of the lamina in the two groups.”.
- “the time of the surgery”
We are grateful to the Reviewer for this suggestion.
In order to clarify the time of surgery, we have added in the Results the Table 4, reporting the duration of each surgical procedures in dogs, cats. In Discussion, we have added the following sentence: “Piezoelectric cut is realized by mechanical vibrations in a range between 60 and 210 μm with a frequency of 29.000 times per second, thus obtaining an efficacy that, although slightly inferior to that of bone saws or burs, has been reported to not increase the overall duration of surgery.”.
- “the power of the instrument in relation to the thickness of the osteotomized bone”
On the basis of this suggestion, we have added the following sentence regarding the power of the instrument: “About the power of the instrument, some presets are available, mainly varying according to the density of the tissue. As a rule, the greater the consistence of the tissue is, the greater the power required. The bone preset involved a power of 16 W and an average frequency of 30 KHz, as reported in the illustrative case 1.”
- “the method of use and the pressure to be exerted on the bone in accordance with Stelzle and Vercellotti”
We have better clarified the concept reported by Stelzle and Vercellotti and other authors as it follows: “Stelzle and Vercellotti reported that a key aspect related to the correct use of the instrument is the pressure load applied on the tip, it largely affecting the overall efficacy and safety of the procedure. They and other authors reported that an excessive pressure load prevents microvibrations of the insert, and the total amount of energy not employed for cutting is converted into heat, ultimately leading to damage the soft tissues and neurovascular structures. As a consequence, irrigation is essential for leading to the cavitation phenomenon and, at the same time, avoiding dangerous overheating.”.
- “the cutting lines and inserts used with their indications”
We have specified these aspects as it follows: “The cutting lines were various in shape and, as reported in Results, the piezoelectric scalpel allowed to perform curved micrometric cuts, which were very useful in different surgical scenarios. About the inserts, the OT7 ones, characterized by an angled large saw tip, are recommended for coarse osteotomies. They are available in 2 different lengths, 10 and 20 mm, to be chosen on the basis of the depth of the bone. As a rule, the longest one is suitable for ventral slots, as reported in the illustrative case 4, or also lumbosacral laminectomy. The tips have some landmarks on their lateral surface, that act as depth meter. The thickness of the tips is 0.55 and 0.6 mm for the short and long one, respectively. In our experience, OT2 insert has been useful in high-precision osteotomies thanks to a 3mm length sharp tip.”.
- “the piezosurgery has established working protocols, discuss if you needed to change them”
In order to answer to this point, we have added the following sentence: “We found all the presets and working protocols as suitable for most of the osteotomies.”.
- “problems related to venous plexus bleeding” “if the piezosurgery is safe also for the venous plexus and not only for the spinal cord”
Thank you for this point. We have clarified this aspect, accordingly: “Regarding the venous plexus, our impression was that, in comparison with rotating instruments previously utilized, the piezoelectric scalpel is no much more aggressive.”.
- “the surgeons’ assessment of osteosclerosis and bone resorption”
In the light of the long-term Results, we have specified that: “The minimal invasiveness of the piezo cut has been also confirmed by the imaging follow-up, where the incidence of the osteosclerosis and bone resorption at the osteotomized surfaces was negligible.”.
- I also suggest to emphasize the importance of the work done, the importance of bone histology and the possible positive consequences on post-surgery inflammation and healing. The title is minimal invasive piezoelectric osteotomy ...
We appreciate the Reviewer for this suggestion, according to which we have emphasized the importance of our work, also highlighting the important role of the histology in proving the minimal invasiveness of the technique, as it follows: “In the present study, whose major strength lies in the high number of treated cases, the non-aggressiveness and safety of the piezoelectric bone scalpel have been also confirmed by the histology, which showed the absence of necrosis along with vitality of the osteocytes and osteoblasts. Similar results have been reported in literature. Piezosurgery has been also reported to interfere in a lesser extent than drills with the initial phases of the bone healing, mainly because stimulating the bone morphogenetic proteins. A better control of iatrogenic inflammatory processes, along with the stimulation of bone remodeling has been also demonstrated.”.
Furthermore, we have rewritten our Conclusions as it follows: “Piezoelectric cut can be considered as a Minimal Invasive Osteotomy technique. In our experience, it has proven to be precise, effective and safe in veterinary craniotomies and laminectomies.
Its main strength lies in the ability to spare the soft tissues and neuro-vascular structures, being moreover free from the risk of thermal injury.”.
- Looking at the bibliography I would suggest to integrate, checking the veterinary articles. For example check the following papers…
Thank you very much also for this advice. We have accordingly added all the suggested articles in our bibliography.
We want to thank once again the kind Reviewer for the precious suggestions which have been paramount for us to improve the overall clarity and quality of the manuscript.

Reviewer 2 Report
In this article, the Authors reported a series of consecutive veterinary minimally invasive piezoelectric osteotomies. Authors retrospectively reviewed Rx and CT scan of each case to evaluate outcome of this technique.
Piezosurgery is a recent tool of the neurosurgeon armamentarium. In literature it is possible to find some preliminary reports about its efficacy in prevent surgery-related complication and its precision in performing osteotomies.
This article unique in its kind is well written, the current literature has been reported in the reference list and the figures well describe the efficacy and safety of piezosurgery. The Authors also reported the histological analysis of spinal cord and dura integrity after piezoelectric osteotomy.
Moreover, the large number of cases reported in this paper and the relative low number of complications, confirmed the minimally invasive nature of piezosurgery.
Author Response
Minimal Invasive Piezoelectric Osteotomy in Neurosurgery: Technic, Applications, and Clinical Outcomes of a Retrospective Case Series
Response to Reviewer 2
Reviewer #2
“In this article, the Authors reported a series of consecutive veterinary minimally invasive piezoelectric osteotomies. Authors retrospectively reviewed Rx and CT scan of each case to evaluate outcome of this technique.
Piezosurgery is a recent tool of the neurosurgeon armamentarium. In literature it is possible to find some preliminary reports about its efficacy in prevent surgery-related complication and its precision in performing osteotomies.
This article unique in its kind is well written, the current literature has been reported in the reference list and the figures well describe the efficacy and safety of piezosurgery. The Authors also reported the histological analysis of spinal cord and dura integrity after piezoelectric osteotomy.
Moreover, the large number of cases reported in this paper and the relative low number of complications, confirmed the minimally invasive nature of piezosurgery”.
We appreciate the comment of the Reviewer very much, and we thank him for the points highlighted.

Reviewer 3 Report
The study has an important number of subjects, but it is structured inadequately. Here there are no inclusion criteria, with results that do not discuss the methods applied. The parameters with which to objectively evaluate the technique are missing (e.g. complications, duration of the procedure, statistical evaluation -which is important in a large number of patients) or are subjective. Therefore the results are not conclusive, as are the non-exhaustive discussions. It is a retrospective work, not a case series: the presentation of 8 cases is excessive. The number and type of photos are not suitable for a retrospective study. In many cases, diagnostic imaging is inadequate or insufficient. The photos show slightly different osteotomies, with clean or more irregular cuts, without being explained or compared (e.g. different types of inserts and tips)
The title speaks of orthopedics patients and clinical outcome, but only neurosurgery is reported the clinical follow up is not reported.
Introduction
line 51, 53, 55: there is no space between the word and the bibliographic citation
line 63: no orthopedic cases are mentioned in the article
Materials and Metods
in the Materials and methods of a retrospective study, the inclusion criteria of the patients must be indicated, the final number is included in the results.
line 66: 292 e 32 in results
line 69-70: it is not veterinary anatomical nomenclature: Cranio Caudal projection, not Antero-Posterior, if orthopedic radiography, Medio-Lateral.
85-86: on what basis was the evaluation made?
87: both group?
88: 2D X-ray?
88: in how many patients was CT or MR performed?
89-90: Have incidence of osteosclerosis and bone resorption been assessed on radiographs? post-operative it is not yet possible to evaluate those alterations, and not in radiology. Have CT been done for evaluation? at what time?
96: follow up how was it done? clinical, with diagnostic imaging? it is not mentioned in the results
Table 1:
there is talk of orthopedic patients, but they are not decrypted
what is meant by cranial?
how do you valuate the mean blood loss if it is not related to the size of the subject?
what do they indicate #?
Results
the results show subjective impressions of the authors. there are no advanced diagnostic images that support. Neither the evaluation methods nor the results about osteosclerosis and bone resorption are mentioned. There is no information about the bone-extracellular matrix. Clinical follow up is not reported.
105: what does clean mean?
116: in how many cases?
Case #1
on what basis was the diagnosis of lumbosacral degenerative stenosis made? from the epidurography of the photo (a) there is a disc disease in situ.
to show the ostectomy site, the photo (d) is sufficient
123: y-o???.
Caso #2
132: what does dorso-lumbar laminectomy mean? hemilaminectomy of T13 and L1 or laminectomy of T13 and/or of L1?
133: postero is not a veterinary nomenclature (dorsal).
Case #3
138: convexity craniotomi: lateral rostro tentorial craniotomy is preferred
photos (f) and (g) are sufficient
Case #4
148 dorsal approaches
the only one photo is sufficient between (b), (c) and (d)
Case #5
the authors speak of thoracic disk extrusion, but the arrow in the photo (a) indicates a disk with mineralization of the nucleus pulposus in situ, in the absence of bone marrow compression. In addition, the radiography shows an epidurography with canalogram, both artifactual complications of a myelography
Excessive number of photos
156: a posterior hemilaminectomy
Case #6
163: caudal cervical spondylopathy is preferred
163: a posterior….
Excessive number of photos: just (c) for radiology. MRI does not confirm the diagnosis: in (d) all the intervertebral discs are well hydrated and the dorsal scan (e) is not useful for showing compression
165: anterior (same consideration for posterior)
Case #7
169: specify better: dorsolumbar hemilaminectomy or dorsal or lumbar laminectomy?
Case #8
177: one again???
178: only in this case the author talks about recovery
181: the author talks about thoracic laminectomy, but the x-rays are only the lumbar portion, and myelography is not seen
Discussion
188: wath means: not interfere with the economy of surgery?
198-199: the concept expressed in the lines 196-197 is repeated
204: various sites, but only for laminectomy/craniotomy
206: wide spectrum, but only for laminectomy/craniotomy
205-208: ampio spettro perché? Solo siti diversi
da quanto esposto in tabella e nei casi, testato non”particularly useful”, ma solo lì.
212: the time parameter was not assessed in the M&M and results. The discussion is based on subjective data
Author Response
Minimal Invasive Piezoelectric Osteotomy in Neurosurgery: Technic, Applications, and Clinical Outcomes of a Retrospective Case Series
Response to Reviewer 3
We want to thank the Reviewer# 3 for his comments and suggestions that have been very precious for us in order to improve the quality and clarity of our manuscript.
Below, we report an itemized, point-by-point response to the Reviewers’ kind remarks.
All the changes in the manuscript have been reported in track change mode ON.
Reviewer #3
- “The study has an important number of subjects, but it is structured inadequately. Here there are no inclusion criteria, with results that do not discuss the methods applied”.
We appreciate the valuable comment of the Reviewer and agree with the need the better clarify the inclusion criteria of the study. We also agree about the fact that a better discussion of the methods applied was necessary. As a consequence,we have indicated clearly the inclusion and exclusion criteria of the study as it follows: “Inclusion criteria involved dogs and cats which required a neurosurgical procedure comprehending a spinal or skull osteotomy because of a traumatic, degenerative or neoplastic lesion. Animals older than 10 years-old or harboring more than a single neurosurgical pathology were excluded.”
Furthermore, we have added the following sentences with the goal to better report the Results of the methods applied:
“A total of 292 dogs and 32 cats were included in the study.
The number of large, medium-sized and small dogs was 97, 121 and 74, respectively. In the cats’ group, 9, 19 and 4 were large, medium-sized and small breed. Average age of dogs was 6 ± 1.41, whereas mean age in cats was 6.6 ± 6. Male/female ratio was 2.8 for dogs and 0.7 for cats. The number of craniotomies and laminectomy/hemilaminectomy was 4 and 305, respectively. In the laminectomy/hemilaminectomy group, T11 -L3 level was interested in 92% of cases. Fifteen were the ventral slots, and C5-C7 segment was involved in 94% of cases.”
“Being the piezoelectric surgery based on the physical principle of the indirect piezo effect and cavitation, its efficacy was practically the same in dogs’ and cats’ laminectomies, regardless by the obvious differences in the thickness of the lamina. On the contrary, the length of the cutting tip may vary in dogs’ and cats’ group.
A rapid learning curve about the use of the piezoelectric scalpel, characterized by a significant improvement of the confidence and dexterity with the instrument, was noted already after the firsts twenty treated cases.”.
“Apart from X-ray, CT scan and MRI were performed postoperatively in 26 and 4 specific cases, respectively, where the animals underwent to surgery because of a spinal traumatic or degenerative pathology.”.
- “The parameters with which to objectively evaluate the technique are missing (e.g. complications, duration of the procedure, statistical evaluation -which is important in a large number of patients) or are subjective. Therefore, the results are not conclusive, as are the non-exhaustive discussions”.
Based on this precious suggestion, we have improved the description of the parameters employed in the evaluation of the technique, and also added some new ones as the duration of the procedure, including the Table 4 in the Results. We have also added the Table 2 and 3, with a statistical analysis aimed to highlight differences in terms of blood loss between the dogs’ and cats’ group. ANOVA test for blood loss in dogs’ and cats’ group was performed.
- “It is a retrospective work, not a case series: the presentation of 8 cases is excessive”.
On the basis of this advice, we have removed the case 5, 6, and 7, thus significantly reducing the number of illustrative cases.
- “The number and type of photos are not suitable for a retrospective study”.
Having decided to not include the case 5, 6, and 7 in the edited version of our manuscript, we have reduced considerably the number of photos. Moreover, according to what kindly and point-by-point indicated by the Reviewer (see below), we have removed some redundant pictures in the most of the illustrative cases or integrated the overall description of the cases with additional radiological images.
- “In many cases, diagnostic imaging is inadequate or insufficient”.
We have realized that some cases were worthy of an important integration with imaging, therefore we have added:
Figure 6 a, g: Pre-operative contrast-enhanced CT scan showing a suspected left parietal meningioma (a). Post-operative CT scan performed at 6-month follow-up showing the sharpness of the edges of the craniotomy. No osteosclerosis or bony resorption were evident (g).
Figure 7 a-d, f: Axial (a), coronal (b), sagittal (c) and 3D (d) cervical CT scan showing a C3-C4 disk extrusion in a 7 years-old dachshund suffering from a progressive tetraplegia.
Figure 8 e: Six-months post-operative ventro-dorsal X-ray showing the site of the laminectomy
- The photos show slightly different osteotomies, with clean or more irregular cuts, without being explained or compared (e.g. different types of inserts and tips)
Being our study aimed at the description of the advantages of the reported technique, we concur with the need to report the different types of inserts and tips used, also making easier a comparison of the different osteotomies. Accordingly, we have:
Changed the figure legend of the illustrative case 1 as it follows: “Figure 4: Dog, 9 years-old, diagnosed with a lumbosacral stenosis. (a) Epidurography showing a severe stenosis. (b) Skeletonization of the lumbosacral segment. Note the incision of the last lumbar and first sacral vertebra, along with the possibility of a further lateral extension (c) Lumbar laminectomy. The thickness of the lamina measure 12 mm. Laminectomy was performed by means of the piezoelectric bone scalpel. An angled large saw tip (OT7, Osteotomy Tips Kit, Mectron Medical Technology, Genoa, Italy) was used. The bone preset involved a frequency of 30 KHz and a power of 16 W. A complete sparing of the underlying dura was observed after the osteotomy. OT7 designated tip also allowed to wide laterally the osteotomy in an easy way after and the dura exposure. Note that this tip allows cuts up to 15 mm of depth.”.
Changed the figure legend of the illustrative case 2 as it follows: “Figure 5. X-ray showing a T13-L1 fracture with hyperkyphosis in a paraplegic dog (a). A safe thoracic piezoelectric laminectomy was obtained with an angled sharp tip (OT2, Osteotomy Tips Kit, Mectron Medical Technology, Genoa, Italy) piezoelectric bone scalpel (b). Dorsal monolateral vertebral stabilization T13-L1 with plate and screws. Noteworthy, piezoelectric scalpel with a different straight tip was used as tapper for the screws placement (c).”
Changed the figure legend of the illustrative case 3 as it follows: “Figure 6. Pre-operative contrast-enhanced CT scan showing a suspected left parietal meningioma (a). Intraoperative pictures showing the initial steps of the craniotomy. The bone flap was cut with piezoelectric scalpel and an angled large saw tip (OT7, Osteotomy Tips Kit, Mectron Medical Technology, Genoa, Italy) (c-d). The dura was completely preserved (e). En bloc removal of meningioma (f). Post-operative CT scan performed at 6-month follow-up showing the sharpness of the edges of the craniotomy. No osteosclerosis or bony resorption were evident (g).”
Changed the figure legend of the illustrative case 4 as it follows: “Figure 7. Axial (a), coronal (b), sagittal (c) and 3D (d) cervical CT scan showing a C3-C4 disk extrusion in a 7 years-old dachshund suffering from a progressive tetraplegia. Ventral slot decompression was obtained by means of long angled large saw tip (OT7, Osteotomy Tips Kit, Mectron Medical Technology, Genoa, Italy) (e).”.
Changed the figure legend of the illustrative case 8 as it follows: “Figure 8. X-ray in latero-lateral (a) and ventro-dorsal (b) projection showing multiple retained lead shots in a cat paraplegic because victim of a firearm trauma. One of the intrathecal lead shot caused a spinal cord contusion and partial compression. A subdural hematoma was also present. Intraoperative pictures showing lumbar spinal cord after decompressive one-level laminectomy performed with piezoelectric scalpel and an angled small saw tip (OT7S-3, Osteotomy Tips Kit, Mectron Medical Technology, Genoa, Italy) (c-d). Six-months post-operative ventro-dorsal X-ray showing the site of the laminectomy (e).”
Added figure 9 a, b: “Fig. 9: 2D (a) and 3D (b) CT scan of an explicative case of dorsal hemilaminectomy performed in a dachshund diagnosed with a spinal cord compression.”
- “The title speaks of orthopedics patients and clinical outcome, but only neurosurgery is reported the clinical follow up is not reported”.
We have changed the title of the article as it follows: “Minimal Invasive Piezoelectric Osteotomy in Neurosurgery: Technic, Applications, and Clinical Outcomes of a Retrospective Case Series”.
Introduction
- “line 51, 53, 55: there is no space between the word and the bibliographic citation”
We have inserted the space at each of the points indicated;
- “line 63: no orthopedic cases are mentioned in the article”;
We have limited to the only neurosurgical cases the results. Accordingly, we have changed the title of the article as it follows: “Minimal Invasive Piezoelectric Osteotomy in Neurosurgery: Technic, Applications, and Clinical Outcomes of a Retrospective Case Series”. Furthermore, we have better classified the procedures, rightly intending all of them as neurosurgical.
Materials and Methods
- “in the Materials and methods of a retrospective study, the inclusion criteria of the patients must be indicated, the final number is included in the results. line 66: 292 e 32 in results”.
We thank you the Reviewer for this suggestion. In order to improve the presentation of the data, we have:
Indicated the overall number of treated patients in the Results section, also eliminating this data from Material and Methods section.
- “line 69-70: it is not veterinary anatomical nomenclature: Cranio Caudal projection, not Antero-Posterior, if orthopedic radiography, Medio-Lateral”.
We have corrected the highlighted inaccuracies.
- “85-86: on what basis was the evaluation made?”
We have clarified this point as it follows: “The irrigation rate of 30 mL/min was arbitrarily decided on the basis of the balance between the cooling effect and a clear vision of the surgical field during the bony work.”. Furthermore, we have also specified some characteristics regarding the instrument’s power as it follows: “Instrument’s power was based on specific presets varying on the basis of the tissue density. The presets ranged between 2.8 and 16 W.”
- “87: both group?”
We have specified that this concept involved both groups.
- “88: 2D X-ray?”
We have changed “2D X-ray” in “X-ray in both the projections”.
- “88: in how many patients was CT or MR performed?”
We have clarified that “Apart from X-ray, CT scan and MRI were performed postoperatively in 26 and 4 specific cases, respectively, where the animals underwent to surgery because of a spinal traumatic or degenerative pathology.”;
- “89-90: Have incidence of osteosclerosis and bone resorption been assessed on radiographs? post-operative it is not yet possible to evaluate those alterations, and not in radiology. Have CT been done for evaluation? at what time?”
We agree with the need to better elucidate this important point, as well as about the fact that the incidence of osteosclerosis cannot be evaluated neither on X-ray, nor in immediate post-op. Accordingly, we have modified the sentence specifying some details as it follows: “The linearity and precision of osteotomies, and the incidence of osteosclerosis and bone resorption in the adjacent segments, were evaluated on imaging studies by two different surgeons.”.
- “96: follow up how was it done? clinical, with diagnostic imaging? it is not mentioned in the results”
We have better explained how the follow-up was done as it follows: “Postoperatively, as a rule, all the animal operated for a skull pathology, underwent to CT scan. Regarding the spinal cases, an X-ray in both the projections was performed by default at six-month follow-up, completed with a CT scan where the owner was available. Nevertheless, further imaging evaluations were decided on a case-by-case basis according to the treated pathology.”. Furthermore, the following aspect have been added in Results: “Apart from X-ray, CT scan and MRI were performed postoperatively in 26 and 4 specific cases, respectively, where the animals underwent to surgery because of a spinal traumatic or degenerative pathology.”.
Table 1:
- “there is talk of orthopedic patients, but they are not decrypted”
We have better classified our procedures, rightly intending all of them as neurosurgical.
- “what is meant by cranial?”
We intend neurosurgical pathology affecting the skull (basically intracranial tumors in the present series); We have changed “cranial” with “skull” in Table 1 and in the text.
- “how do you valuate the mean blood loss if it is not related to the size of the subject?”
Thank also for this point. We measured intraoperatively the blood loss trough the evaluation of the fluids presents in the suction receptals. However, it was assumed to be a broad evaluation of the real blood loss, this last certainly overestimated by the continuous irrigation. This point has been clarified in Material and Methods section as it follows:“In order to have a rough idea about the invasiveness of the piezoelectric cut, average blood loss was measured in all animals. The blood loss was calculated as the difference between the known amount of irrigation liquids attached to the peristaltic pump, and those present in the suction receptals.”.
- “what do they indicate #?”
The symbol # universally indicates the term "number". However, we have changed in the manuscript the symbol #, which may result misleading, with the symbol “N°”.
Results
- “the results show subjective impressions of the authors”.
We completely agree with the Reviewer with the need to better clarify those criteria used to objectively assess the rate of post-operative osteosclerosis of the cut surface. This to avoid that the interpretation of the results may appear as purely subjective. For this reason, we have explained our outcome measures in Materials and Methods section as it follows: “The criteria adopted to objectively evaluate the long-term incidence of osteosclerosis and bone resorption at the level of the cut surfaces were derived by a modification of that reported by Bolm-Audorff and colleagues for the lumbar spine. In particular, both on X-ray and CT scan, we considered as osteosclerotic a cut surfaces where the thickness of cortical bone extended into the cancellous bone by > 2 mm.”.
- “there are no advanced diagnostic images that support”.
The aforementioned criteria by Bolm-Audorff were chosen by our group for 2 main reasons, namely the high reliability reported in literature, and their easiness of application which involves X-ray radiographs. However, as explained, we used also CT scan when possible.
- “Neither the evaluation methods nor the results about osteosclerosis and bone resorption are mentioned”.
As explained before, we have added the following sentence in the Materials and Methods section: “The criteria adopted to objectively evaluate the long-term incidence of osteosclerosis and bone resorption at the level of the cut surfaces were derived by a modification of that reported by Bolm-Audorff and colleagues for the lumbar spine37. In particular, both on X-ray and CT scan, we considered as osteosclerotic a cut surfaces where the thickness of cortical bone extended into the cancellous bone by > 2 mm.” with the aim to better define which were our evaluation methods.
- “There is no information about the bone-extracellular matrix”.
Our study was clinical, observational and retrospective. In particular, it was based on an animal cohort surgically managed with a curative intention. This aspect unfortunately precluded the possibility to perform a long-term histological follow-up evaluation aimed at the assessment of the bone-extracellular matrix, as well as cellular viability.
- “Clinical follow up is not reported.”
Thank you also for this point. We would like to point out that our clinical follow up focused on the results of piezosurgery rather than the pathology, whose clinical evolution was assumed to be independent by the surgical technique employed.
- “105: what does clean mean?”
With the term “clean”, we basically referred to a surgical field free form blood and debris. In order to avoid misleading, we have changed the term “clean” as it follows:
Case N° 1 “The procedure was fast and bloodless…”
Results: “Operative field resulted free from blood and debris in all cases…”
- “116: in how many cases?”
We have clarified that the post-mortem cases where the spinal cord was harvested were 17 with the following sentence (Results): “Harvested spinal cord in 17 post-mortem cases allowed to appreciate the anatomical integrity of both the dura and the neural tissue”.
Case #1
- “on what basis was the diagnosis of lumbosacral degenerative stenosis made? from the epidurography of the photo (a) there is a disc disease in situ”.
We have specified that the diagnosis was made on basis of the epidurography (Fig. 4 a).
- “to show the ostectomy site, the photo (d) is sufficient”
The kind suggestion of the Reviewer has been followed and the Fig. 4 “b” and “e” have been removed. Fig. 4 c has been maintained to better show the linearity of the piezo cut.
- “123: y-o???”.
The age of the dog (9 years-old) has been specified in the figure legend (Fig. 4)
Case #2
- “132: what does dorso-lumbar laminectomy mean? hemilaminectomy of T13 and L1 or laminectomy of T13 and/or of L1?”
We apologize for this inaccuracy. We have changed the sentence: “She underwent to a dorso-lumbar laminectomy with a postero-lateral internal arthrodesis involving screws and plate” in the following sentence: “The dog underwent to a thoracic laminectomy with a dorsal monolateral vertebral stabilization T13-L1 with plate and screws”.
- “133: postero is not a veterinary nomenclature (dorsal).”
We thank you also for this clarification. We have changed the term “postero” with “dorsal” in the description of the case.
Case #3
- “138: convexity craniotomi: lateral rostro tentorial craniotomy is preferred photos (f) and (g) are sufficient”
Thank you also for this point. The term “convexity craniotomy” has been changed in “lateral rostro tentorial” craniotomy. Furthermore, Fig. 6 “a”, “c” and “h” have been removed.
Case #4
- “148dorsal approaches the only one photo is sufficient between (b), (c) and (d)”
The term “posterior approaches” has been changed “dorsal approaches”. Fig. 7 “c” and “d” have been removed.
Case #5
- “the authors speak of thoracic disk extrusion, but the arrow in the photo (a) indicates a disk with mineralization of the nucleus pulposus in situ, in the absence of bone marrow compression. In addition, the radiography shows an epidurography with canalogram, both artifactual complications of a myelography
- Excessive number of photos
- 156: a posterior hemilaminectomy”
Based on the kind previous suggestions of the Reviewer (“It is a retrospective work, not a case series: the presentation of 8 cases is excessive”), we have removed the illustrative case 5.
Case #6
- “163: caudal cervical spondylopathy is preferred
- 163: a posterior….
- Excessive number of photos: just (c) for radiology. MRI does not confirm the diagnosis: in (d) all the intervertebral discs are well hydrated and the dorsal scan (e) is not useful for showing compression
- 165: anterior (same consideration for posterior)”
Based on the kind previous suggestions of the Reviewer (“It is a retrospective work, not a case series: the presentation of 8 cases is excessive”), we have removed the illustrative case 6.
Case #7
- “169: “specify better: dorsolumbar hemilaminectomy or dorsal or lumbar laminectomy?”
Based on the kind previous suggestions of the Reviewer (“It is a retrospective work, not a case series: the presentation of 8 cases is excessive”), we have removed the illustrative case 7.
Case #8
- 177: “one again???”
We have decided to leave this case because representative, having removed instead the case 5, 6 and 7.
- 178: “only in this case the author talks about recovery”
We thank you the Reviewer once again for this comment. Accordingly, we have specified the recovery in the other illustrative cases as it follows:
Case 1: “Two weeks after surgery, the walk significantly improved”.
Case 2: “At the six-month follow-up the dog partially recovered”.
Case 3: we already reported this information “Tumor was completely removed, and recovery was complete without complications”.
Case 4: “At the six-month follow-up, the motor deficits significantly improved, and no signs of instability were documented.”
- 181: “the author talks about thoracic laminectomy, but the x-rays are only the lumbar portion, and myelography is not seen”
We regret for this typo error and we thank the reviewer for having highlighted it. We have changed “thoracic” in “lumbar”.
Discussion
- 188: “what means: not interfere with the economy of surgery?”
In order the clarify the concept, we have changed the sentence “…non interferes with the economy of surgery” with “not increases the overall duration of surgery”, being our intention to stress the concept that the technique has been not time-consuming in our experience.
- 198-199: “the concept expressed in the lines 196-197 is repeated”
Thank you also for this consideration. We have realized that the concept was repeated and, accordingly, we have changed the sentence: “A key aspect related to the correct use of the instrument is the pressure load applied on the tip, largely affecting the efficacy and safety of the procedure. Stelzle and Vercellotti reported that the efficacy of the piezoelectric cut is linked to the pressure applied on both the instrument, and the insert. An excessive pressure load prevents microvibrations of the insert, and the total amount of energy not employed for cutting is converted into heat, ultimately leading to damage soft tissue and neurovascular structures.” with the following sentence: “Stelzle and Vercellotti reported that a key aspect related to the correct use of the instrument is the pressure load applied on the tip, it largely affecting the overall efficacy and safety of the procedure. They and other authors reported that an excessive pressure load prevents microvibrations of the insert, and the total amount of energy not employed for cutting is converted into heat, ultimately leading to damage the soft tissues and neurovascular structures”.
- 204: “various sites, but only for laminectomy/craniotomy”
Rightly, having limited our study to the skull and spinal pathologies, we agree with the Reviewer with the need to limit the concept to the same sites. Therefore, we have changed the sentence “The present study has been conducted on dogs and cats affected by different pathologies in various anatomical sites.” with the following sentence: “The present study has been conducted on dogs and cats affected by skull and spinal pathologies”.
- 206: “wide spectrum, but only for laminectomy/craniotomy”
We have changed the sentence: “This aspect allowed us to test the effectiveness, invasiveness and safety of the instrument in a wide spectrum of surgical situations.” with the following sentence: “This aspect allowed us to test the effectiveness, invasiveness and safety of the instrument, especially in laminectomies and craniotomies.”
- 212: “the time parameter was not assessed in the M&M and results. The discussionis based on subjective data”
As explained before, we fully agree with the kind Reviewer with the need clarify which were our objective criteria to assess the rate of post-operative osteosclerosis of the cut surface. As a consequence, Materials and Methods section, we have added the following sentence: “The criteria adopted to objectively evaluate the long-term incidence of osteosclerosis and bone resorption at the level of the cut surfaces were derived by a modification of that reported by Bolm-Audorff and colleagues for the lumbar spine. In particular, both on X-ray and CT scan, we considered as osteosclerotic a cut surfaces where the thickness of cortical bone extended into the cancellous bone by > 2 mm.”
We want to thank once again the kind Reviewer for the precious suggestions which have been paramount for us to improve the overall clarity and quality of the manuscript.

Reviewer 4 Report
Line 90 specify how is possible that osteosclerosys is visible in the postsurgical control
Line 133 replace postero lateral artrodesis with lateral fixation with screws and plate
Line 135 Figure 5 c) the same correction
Line 138 replace where
Line 149 figure a: ct scn shows erroneously a lumbar vertebra
Line 156 replace posterior with dorsal
Line 165 Figure 9. the pictures a,b,c show spinal cord compression at intervertebral space C5/C6 while picture d shows spinal cord compression at intervertebral space C4/C5
Line 163 replace posterior with dorsal
Line 176 replace fragment of bullet with lead shot
Line 179 figure 11 the same correction
Author Response
Minimal Invasive Piezoelectric Osteotomy in Neurosurgery and Orthopedics: Technic, Applications, and Clinical Outcomes of a Retrospective Case Series
Response to Reviewer 4
We want to thank the Reviewer# 4 for his comments and suggestions that have been very precious for us in order to improve the quality and clarity of our manuscript.
Below, we report an itemized, point-by-point response to the Reviewers’ kind remarks.
All the changes in the manuscript have been reported in track change mode ON.
Reviewer #4
- Line 90 “specify how is possible that osteosclerosys is visible in the postsurgical control”
We thank the Reviewer for this suggestion. We also apologize for the misleading sentence. We have clarified the modalities of our follow-up as it follows: “The linearity and precision of osteotomies, and the incidence of osteosclerosis and bone resorption in the adjacent segments, were evaluated on imaging studies by two different surgeons.”.
- Line 133 “replace postero lateral artrodesis with lateral fixation with screws and plate”
Thank you also for this point. We have changed, also in according to what suggested by other Reviewers, the sentence as it follows: “The dog underwent to a thoracic laminectomy with a dorsal monolateral vertebral stabilization T13-L1 with plate and screws (Fig 5)”.
- Line 135 “Figure 5 c) the same correction”
We have made the same correction in Figure 5 c): “Dorsal monolateral vertebral stabilization T13-L1 with plate and screws”
- Line 138 “replace where”
We apologize for this typo mistake. We have changed “where” with “with”.
- Line 149 “figure a: ct scan shows erroneously a lumbar vertebra”
We have replaced figure a with a different one.
- Line 156 “replace posterior with dorsal”
Thank you also for this suggestion. However, by the reason of the fact that other Reviewers suggested to reduce the number of cases, we have removed the illustrative case 5, 6 and 7 from the manuscript.
- Line 163 “replace posterior with dorsal.” Line 165 “Figure 9. the pictures a,b,c show spinal cord compression at intervertebral space C5/C6 while picture d shows spinal cord compression at intervertebral space C4/C5”
According to the aforementioned reasons, we have removed the illustrative case 6 from the manuscript.
- Line 176 “replace fragment of bullet with lead shot” Line 179 “figure 11 the same correction”
In both the parts kindly indicated by the Reviewer, we have replaced “fragment of bullet” with “lead shot”.
We want to thank once again the kind Reviewer for the precious suggestions which have been paramount for us to improve the overall clarity and quality of the manuscript.

Round 2
Reviewer 1 Report
Dear Authors the new version of the paper shows that it has changed considerably from the previous one. However, some concepts still have to be clarified:
lines 42-59
I wish the piezoelectric principles were better explained. Osteotomy is not based only on the phenomenon of cavitation.
Osteotomy is performed thanks to the mechanical effect of ultrasound at certain frequencies.
This mechanical effect is increased by the cavitation effect which is generated by the contact with the dispensing solution. This also allows to do bloodless surgeries. Therefore I would invert the order of the concepts expressed in the introduction and better explain these principles.
line 171
in the table 4 change "anterior cervical spine" with "cranial cervical spine" and "posterior cervical spine" with "caudal cervical spin"
line 220
change "dorsal monolateral vertebral stabilization" in "monolateral vertebral stabilization"
line 242
In the text that refers to Case 4 a compression C3-C4 is explained, however the images refer to a C2-C3 compression. As it's evident that in the picture the C2 vertebra is shown, it's logical to think that there is something that does not coincide.
Discussion
At line 185 it was written that "In two cases of trauma, mechanical damage attributable to the use of the piezoelectric bone scalpel (0.6%) was observed in the group of dogs. They consisted of a dural tear and an epidural hematoma."
It is necessary to discuss this part and give an interpretation on why this happened, and what needs to be done to prevent this iatrogenic damage from happening.
Author Response
Minimal Invasive Piezoelectric Osteotomy in Neurosurgery: Technic, Applications, and Clinical Outcomes of a Retrospective Case Series
Response to Reviewer
We want to thank the Reviewer# 1 for his further comments and suggestions that have been fundamental for us in order to improve even more the quality of our manuscript.
Below, we report an itemized, point-by-point response to the Reviewers’ kind remarks.
All the changes in the manuscript have been reported in track change mode ON.
Reviewer #1
- “lines 42-59. I wish the piezoelectric principles were better explained. Osteotomy is not based only on the phenomenon of cavitation.
Osteotomy is performed thanks to the mechanical effect of ultrasound at certain frequencies.
This mechanical effect is increased by the cavitation effect which is generated by the contact with the dispensing solution. This also allows to do bloodless surgeries. Therefore I would invert the order of the concepts expressed in the introduction and better explain these principles.”
We thank the Reviewer for this important and right observation, according to which we have better explained the piezoelectric principles, inverting the order of the concepts expressed in Introduction, as it follows: “Piezoelectric osteotomy is based upon the mechanical effect of ultrasound, which is increased by physical phenomenon of cavitation. Microvibrations of the ultrasonic frequency are linear in shape and micrometric (60–210 μm). They allow to selectively cut the bone without significant damage to the underlying soft and neurovascular tissues. Oscillation frequency ranges between 25 and 30 kHz, the same frequencies at which soft tissues, dura mater, nerves and blood vessels oscillate at the same time being spared. On the other hand, the cavitation phenomenon involves the formation of vapor-filled cavities secondary to the abrupt and rapid changes of pressure in a liquid. The input coming from a high-pressure source causes collapse of these cavities and generate a shock wave which propagates in the tissue. In turn, the shock wave causes cell detachment from substrate, significant changes in focal adhesion and biomechanics, ultimately inducing a mechanical cut on the mineralized tissue. In piezoelectric surgery, the changes of pressure are generated by a spatial deformation of the piezoelectric crystals when they are subject to an electric field having a given ultrasonic frequency. This mechanism is also called indirect piezo effect. The resulting oscillations are then amplified and transferred onto an instrument tip. Piezoelectric osteotomy is also considered as a “heat-free” technique.
In fact, although the cavitation effects can increase the temperature of the neighboring tissues, the dissipation of the heat coming from the concomitant irrigation, practically leads to avoid any thermal injury. It also promotes the absence of blood and debris in the surgical field”
- “line 171. In the table 4 change "anterior cervical spine" with "cranial cervical spine" and "posterior cervical spine" with "caudal cervical spine"
We have replaced “anterior cervical spine" with "cranial cervical spine" and "posterior cervical spine” with “caudal cervical spine”.
- “line 220. Change "dorsal monolateral vertebral stabilization" in "monolateral vertebral stabilization"
We have changed “dorsal monolateral vertebral stabilization” in “monolateral vertebral stabilization”.
- “line 242. In the text that refers to Case 4 a compression C3-C4 is explained, however the images refer to a C2-C3 compression. As it's evident that in the picture the C2 vertebra is shown, it's logical to think that there is something that does not coincide.”
Thanks also for these important observation. We apologize for this error.
We have corrected the mistake replacing “C3-C4” with “C2-C3”.
Discussion
- “At line 185 it was written that "In two cases of trauma, mechanical damage attributable to the use of the piezoelectric bone scalpel (0.6%) was observed in the group of dogs. They consisted of a dural tear and an epidural hematoma." It is necessary to discuss this part and give an interpretation on why this happened, and what needs to be done to prevent this iatrogenic damage from happening.
Thanks also for this point. We agree with the need to discuss the complications observed and the precautions needed to avoid them. Accordingly, we have added the following sentence in Discussion: “The two complications reported in the present series, namely a dural tear and an epidural hematoma, were both attributable to an accidental slippage of the instrument, with a consequent mechanical damage caused by the tip of the scalpel. The avoiding of an excessive pressure on the instrument handpiece is important also to avoid these potential complications.”
We want to thank once again the kind Reviewer for the precious suggestions which have been paramount for us to improve the overall clarity and quality of the manuscript.

Reviewer 3 Report
all my indications have been correct. no other comments
Author Response
We want to thank the Reviewer# 3 for his comments and suggestions that have been very precious for us in order to improve the quality and clarity of our manuscript.
Sabino Luzzi and all the authors
